

# Robust historical evapotranspiration trends across climate regimes

Sanaa Hobeichi[1,2], Gab Abramowitz[1,2], and Jason Evans[1,2]

[1] Climate Change Research Centre, UNSW Sydney, NSW 2052, Australia.

[2] ARC Centre of Excellence for Climate Extremes, UNSW Sydney, NSW 2052, Australia.

Correspondence: Sanaa Hobeichi (s.hobeichi@unsw.edu.au)

## Abstract

Evapotranspiration (ET) links the hydrological, energy, and carbon cycle on the land surface. Quantifying ET and its spatiotemporal changes is also key to understanding climate extremes such as droughts, heatwaves and flooding. Regional ET estimates require reliable observationally-based gridded ET datasets, and while many have been developed using physically-based, empirically-based and hybrid techniques, their efficacy, and particularly the efficacy of their uncertainty estimates, is difficult to verify. In this work, we extend the methodology used in Hobeichi et al. (2018) to derive a new version of the Derived Optimal Linear Combination Evapotranspiration (DOLCE) product, with observationally constrained spatiotemporally varying uncertainty estimates, higher spatial resolution, more constituent products and extended temporal reach (1980-2018). After successful evaluation of the efficacy of these uncertainty estimates out-of-sample, we derive novel ET climatology clusters for the land surface, based on the magnitude and variability of ET at each location. The verified uncertainty estimates and extended time period then allow us to examine the robustness of historical trends spatially and in each of these six ET climatology clusters. We find that despite robust decreasing ET trends in some regions, these do not correlate with behavioural ET clusters. Each cluster, and the vast majority of the Earth's surface, show clear robust increases in ET over the recent historical period.

## 1. Introduction

Understanding the spatiotemporal variability of evapotranspiration (ET) is a critical part of understanding the processes that lead to high impact weather phenomena, such as droughts (Han et al. 2018; Montano et al. 2015; Sheffield, Wood, and Roderick 2012; Teuling et al. 2013), heatwaves (Teuling 2018; Ukkola et al. 2018) and flooding (Dawdy, Lichty, and Bergmann 1972; Sharma, Wasko, and Lettenmaier 2018). Several global gridded ET datasets have been developed, using physical schemes with different scopes and complexity (see Fisher and Koven, 2020) and empirical techniques including machine-learning algorithms (Hamed Alemohammad et al. 2017; Jung et al. 2010, 2019), typically incorporating a range of remote sensing inputs. Recently, ET datasets derived with a hybrid approach





have been recognised for their potential to outperform single source datasets (Ershadi et al. 2014; Feng
et al. 2016; McCabe et al. 2016; Pan et al. 2020).
While most observational products are global (or near global) in their spatial extent, and typically
available with a monthly time step, different products are constrained by very different types of
observations, and vary significantly in their treatment of uncertainty. As detailed below when describing
the datasets we use here, 'physically-based' approaches, use equations that represent different
physical, chemical, and biological processes and incorporate satellite-based atmospheric forcing, and
parameterization of land surface characteristics, while 'empirical' approaches integrate ground-based
measurements of ET together with satellite data and ground-based measurements of vegetation
characteristics and land surface parameters. These differences result in a diverse group of products and
estimates, but it is their approach to deriving uncertainty estimates that is arguably more important.
Very few datasets provide uncertainty estimates associated with the ET flux, these include datasets
described in Bodesheim et al. (2018) and Jung et al. (2019). In Bodesheim et al. (2018), monthly
uncertainty estimates are computed from the standard deviation of the half-hourly ET values that were
used to derive monthly ET averages. Jung et al. (2019) provide an ensemble of global ET estimates,
deviations from the ensemble median are used to derive ET uncertainties. In both cases, uncertainties
do not reflect the actual deviation from the measured ET at site locations. Without well calibrated
uncertainty estimates we are unable to tell whether an identified property of any given data set, such as
a trend or a proportion of the surface energy or water budget, is robust, rather than a result of bias or
stochastic uncertainty.
ET trends computed from different approaches (i.e. physical and empirical) show general agreement at
the global scale, and indicate that ET has increased since early 1980s (Miralles et al. 2014; Pan et al.
2020; Zhang et al. 2016). However, different ET products exhibit considerable disparities in regional and
continental ET trends. For instance, Miralles et al. (2014) detected upward ET trends in GLEAM (Global
Land Evaporation Amsterdam Model; Miralles et al. 2011) in the northern latitudes caused by vegetation
greening. In water limited regions, they found that ET is characterised by a multidecadal variability that
follows ENSO dynamics, mainly in eastern and central Australia, southern Africa and eastern South
America. In comparison, ET trends estimated from the observation-driven Penman-Monteith-Leuning
(PML; Zhang et al. 2016) model show increasing ET since 1980 in the northern latitudes, arid regions in
northern Africa, and northern and eastern Amazon. On the other hand, PML exhibits negative trends in
southern South America and western United States. More recently, Pan et al. (2020) found that ET
trends exhibited by a range of empirical and physical based estimates disagree in the direction of trend
in the Amazon basin and many arid and semi-arid regions. Without incorporating uncertainties in ET
estimates in the analysis of trends, it becomes difficult to assess the reliability of the established trends.
The gridded ET product derivation technique implemented by Hobeichi et al. (2018) offers the potential
for robust out-of-sample testing of its uncertainty estimates, as well as several other advantages over
other techniques. Like other merging approaches it offers the potential to minimise the eccentricities or
biases of any one product, by averaging them (in this case using weights). However, unlike several other
merging techniques (Mueller et al. 2013; Paca et al. 2019; Rodell et al. 2015; Stephens et al. 2012) it
accounts for performance differences between parent estimates using in-situ data as the observational





constraint, rather than assigning weights based on the ability to match another gridded dataset that is
deemed more reliable, or the ensemble mean of a selection of datasets (Munier et al. 2014; Sahoo et al.
2011; Wan et al. 2015; Zhang et al. 2018). The efficacy of using in-situ measurements for constraining
much larger scale gridded estimates has also been shown explicitly (Hobeichi et al. 2018; Hobeichi,
Abramowitz, Contractor, et al. 2020). Next, most available merging techniques do not account for
dependence between parent estimates, where redundant information in different parent products is
likely to bias the hybrid estimate (Abramowitz et al. 2019; Herger et al. 2018). Finally, and perhaps most
important for this work, the technique calculates global spatially and temporally varying uncertainty
estimates that are observationally-based, in that they are based on the discrepancy between the hybrid
ET estimate and in-situ data. Aside from being more defensible than simply taking the spread of the
parent products around their mean (e.g. Pan et al., 2012, Zhang et al., 2018), this approach also allows
for out-of-sample testing, by leaving some sites out of the derivation of the hybrid product and its
uncertainty, and then using them to test its accuracy.
Despite these advantages, out-of-sample testing of uncertainty estimates was not explored by Hobeichi
et al (2018), and the short temporal availability of the DOLCE product (2000 – 2009) limited its
application, particularly in examining historical trends. While different subsets of parent products were
used over different regions to expand the spatial coverage of DOLCE, the possibility of different product
subsets in different time periods to extend its temporal reach was not explored. Additionally, since the
development of DOLCE, four of its six parent datasets (Jung et al. 2010; Martens et al. 2016; Miralles et
al. 2011; Mu, Zhao, and Running 2011; Zhang et al. 2016) have been improved and several new global ET
datasets have been developed (Balsamo et al. 2015; Bodesheim et al. 2018; Jung et al. 2019). Most of
these are available at a higher spatial resolution than the original 0.5° in DOLCE and cover different
subsets of the period 1980 – 2018, with at least two available for every year during this period (Table1).
In this paper we amend these shortcomings and explore some of the insights that the new version of
DOLCE offers, in particular focusing on the temporal trends in ET in different regions, and the
assessment of robustness of trends that well calibrated uncertainty estimates afford. Roughly in order,
we detail below: (1) how we update the DOLCE product with new parent datasets, extended temporal
coverage and higher spatial resolution; (2) how the improved product compares to its previous version
and other existing ET estimates from the literature; (3) the efficacy of uncertainty estimates, in
particular whether or not they are overconfident; (4) an exploration of historical trends in ET using the
extended temporal coverage, and how the uncertainty estimates allow us to examine the robustness of
these trends (5) behavioural ET clusters that describe ET based climate regimes, as a mean to
understand the distribution of trends we find.

## 2. Data and Methods

To derive a new version of DOLCE, we combine 11 available global gridded ET datasets using the same
merging technique as in DOLCE V1. This technique derives a linear combination of the participating ET
datasets based on their ability to match in-situ observations while also accounting for their error
dependency. While we acknowledge the obvious spatial mismatch between gridded and in-situ data, we
refer readers to Hobeichi et al (2018) where it was shown that in-situ observations do contain useful



information about grid scale fluxes, using out-of-sample testing in a similar framework to the one we
present here.
Our aim is to increase the time coverage and spatial resolution of DOLCE V1, as well as examine
strategies to improve the effectiveness of the weighting strategy. Below we detail newly available global
datasets that allow us to derive DOLCE V2 at 0.25° spatial resolution, and an improved collection of in-
situ constraining data. We then briefly revisit the weighting and uncertainty estimation approach, before
describing our tiering approach to extending the temporal reach of DOLCE V2. Finally, we examine
alternative clustering and bias-correction approaches to improve the out-of-sample performance of the
weighting technique.
Throughout the paper, we use the two terms evapotranspiration (ET) and latent heat (LE)
interchangeably, and the unit $W\ m^{-2}$ for heat fluxes and $mm\ year^{-1}$ for the water flux equivalent. For
reference: $1\ W\ m^{-2} = 12.86\ mm\ year^{-1}$. As above, we refer to the product from Hobeichi et al (2018)
as DOLCE V1 and the new product we are deriving as DOLCE V2 or DOLCE V2.1 .

## 2.1 Data

### 2.1.1 Global ET datasets:

DOLCE V1 was derived from 6 global ET datasets: MPIBGC (Jung et al. 2010), GLEAM v2a, GLEAM v2b,
GLEAM v3a, MOD16 (Mu et al., 2011) and PML. In DOLCE V2, we keep both MOD16 and PML datasets,
substitute the GLEAM products with their improved and latest versions (i.e. GLEAM3.3A and
GLEAM3.3B; Martens et al., 2016, 2017), and replace MPIBGC with newly developed empirical ET
datasets from the Max Planck Institute for Biogeochemistry: BACI and two ET estimates from the
FLUXCOM projects. Additionally, we incorporate a recently published dataset ERA5-Land and three
newly available ET datasets PLSH, SEBS and SRB-GEWEX. We provide a brief description of these
datasets below, with URLs and download dates shown in supplementary Table S2.
Biosphere Atmosphere Change Index (BACI; Bodesheim et al., 2018): The dataset is derived by upscaling
diurnal cycles of ET and other land-Atmosphere fluxes from a large set of FLUXNET sites based on a
random forest regression framework. It uses seasonal vegetation variables and indices from MODIS
satellites, and meteorological data either measured at the flux tower sites or retrieved from the ERA-
Interim data.
ERA5-Land (Balsamo et al. 2015): A global land surface reanalysis dataset that has been developed by
rerunning the land component of the ECMWF ERA5 climate reanalysis with a series of improvements
(mainly higher temporal frequency and spatial resolution) that makes it more reliable for land
applications. ERA5-Land is produced under a single simulation that uses adjusted atmospheric inputs from
ERA5 atmospheric variables without being coupled to the atmospheric module of ERA5.
FLUXCOM (Jung et al. 2019): An empirical upscaling of observations from 224 flux tower sites using
machine learning methods. The full FLUXCOM product includes 63 global ET datasets that have been
produced using two different setups, a remote sensing (RS) setup and a remote sensing + meteorological



(MET) setup. The development of the global datasets incorporates 9 machine learning techniques, 4 global
meteorological datasets (used only with the MET setup), 3 correction methods for energy imbalance at
the flux tower sites and MODIS remote sensing input. In DOLCE V2, we include one dataset from each
setup, that we refer to as FLUXCOM-RS (from the RS setup) and FLUXCOM-MET (from the MET setup). To
choose the two datasets we analysed the pair-wise error correlations of all the products against in-situ
flux tower and selected the two that had the lowest pair-wise error correlation (and so were deemed least
dependent).
Process-based Land Surface Evapotranspiration/Heat Fluxes algorithm (PLSH; Zhang et al., 2015):
Terrestrial ET is derived using an improved NDVI-based Penman-Monteith algorithm originally developed in
(Zhang et al. 2010). ET is regulated by a set of geophysical data from GIMMS and Vegetation Index and
Phenology  along with radiative data from World Climate Research Programme/Global Energy and
Water-Cycle Experiment (WCRP/GEWEX) Surface Radiation Budget (SRB) and CERES along with other
meteorological observations data from the NCEP/DOE AMIP-II Reanalysis (NCEP2; Kanamitsu et al.,
176   2002).

Surface Energy Balance System  (SEBS; Chen et al., 2019; Su, 2002): ET estimates are produced with the
revised Surface Energy Balance System (SEBS) algorithm in Chen et al. (2013; 2019). It uses
meteorological observations, ground heat flux, net radiation and canopy measurements collected from
flux tower sites, and NDVI and emissivity data from MODIS.
Surface Radiation Budget (SRB)-GEWEX (Vinukollu et al. 2011): ET is estimated based on the Penman-
Monteith equation. Input data sets include remote sensing data from AVHRR and MODIS,
meteorological data derived from the Variable Infiltration Capacity (VIC; Liang et al., 1994) land surface
model forced by PGF and radiative data from the NASA Global Energy and Water Exchanges (GEWEX)
Surface Radiation Budget Project (Stackhouse Jr et al. 2011).
It is clear that different parent datasets share forcing, parameterisations, and physical and empirical
assumptions. Therefore, they do not constitute entirely independent estimates. Furthermore, their error
correlation (when compared with data from 254 sites – details on these below), which can be used as a
measure of their dependence (Bishop and Abramowitz 2013) is high (Fig. S2, correlation > 0.5),
reinforcing the potential for benefit using a weighting approach that can account for this redundancy.
Part of the high correlation is of course due to spatial heterogeneity and the scale mismatch between in-
situ and gridded data sets – individual site locations within a grid cell are likely biased with respect to the
(unknown) true grid cell averaged flux. While it might appear that a weighting approach that accounts
for error correlations between parent data sets might be in danger of overfitting to error correlation
resulting from spatial heterogeneity, we have two mechanisms that ensure this is not a concern for our
final product. First, weights for each product are constructed over very large spatiotemporal domains, so
that the (assumed stochastic) biases of individual sites relative to grid cell values are unlikely to
influence weights over a large sample. Second, and more categorical, all results here are presented out-
of-sample, so that any overfitting will degrade, rather than improve the results we present. More detail
on this is presented below.




Given that most of the parent datasets provide ET information at a 0.25° or finer spatial resolution
(Table 1), it is possible to enhance the resolution of DOLCE from 0.5° to 0.25°. All the parent datasets are
resampled from their original spatial resolution to a common 0.25° grid using the nearest-neighbour
resampling method, and aggregated to monthly temporal scale before implementing the weighting
technique.


**2.1.2 Flux tower data**
We use flux tower observations from a range of networks including Ameriflux (ameriflux.lbl.gov), The
Atmospheric Radiation Measurement (ARM; arm.gov), AsiaFlux (asiaflux.net), European Fluxes Database
(europe-fluxdata.eu), Fluxnet 2015, LaThuile Free Fair Use (fluxnet.fluxdata.org), Oak Ridge data
repository (daac.ornl.gov), OzFlux (ozflux.org.au) and through communication with individual site
principal investigators (PI). Particular efforts were made to establish connections with PIs in regions
where ET observations are scarce, including all areas outside North America, Europe and Australia,
particularly the MENA regions, Siberia, Central Africa and the Amazon basin. Our efforts and
communications with many PIs unfortunately failed to incorporate flux data from some of these regions
(excepting those that are already available from the cited networks). Before the quality control process
detailed below, we had obtained data from 366 flux tower sites.

The raw data consists of a composite of half hourly, daily and monthly records. We compute daily
averages from half-hourly records for days where at least 80% of half-hourly LE records are available.
Subsequently, we compute monthly averages from daily records for months where at least 80% of daily
LE records are available. In DOLCE V1 we applied a less strict quality control on the observational data in
which up to 50% of gap filling was allowed. The reason was that DOLCE V1 incorporated much fewer
observational data – sourced from Fluxnet 2015 and LaThuile Free Fair use only. In order to retain
enough observational data to constrain the weighting, it was necessary to make a trade-off between the
quality and the quantity of the data.

We also apply energy balance corrections to the monthly LE at all sites where monthly averages of the
other variables of the surface energy budget - net radiation ($R_n$), ground heat flux ($G$), and sensible
heat flux ($H$) - are available with the same high quality (quality flag > 80%). Corrections are carried out
independently for every monthly record. Where any of the other components of the energy budgets are
absent, latent heat measurements are used directly. The energy balance correction is applied as a
Bowen Ratio (BR) based correction that distributes the energy budget residuals among $H$ and LE in such
a way that their ratio is conserved. This is done under pre-defined constraints that disallow large
changes to be applied to LE. As a result of this, if the original monthly LE and the corrected LE ($LE_{cor}$)
satisfy:
$$\begin{cases} \frac{LE_{cor}}{LE} \epsilon \left[\frac{1}{2} - 2\right], & where \ LE \leq 30 \ W \ m^{-2} \\ LE_{cor} - LE \leq 20 \ W \ m^{-2}, & where \ LE \geq 30 \ W \ m^{-2} \end{cases}$$





we accept the BR correction and use the corrected $LE_{cor}$ values. In DOLCE V1, we did not set a threshold
for LE adjustments, which resulted in LE being changed drastically in a few sites to offset errors in the
other energy balance components. If the BR correction does not meet the above criterion, we reject the
correction and try using a residual correction, which simply calculates LE as the residual term in the
energy balance equation, i.e. $LE_{cor} = R_n - H - G$ . Similarly, we reject the residual correction if the
relation between LE and $LE_{cor}$ above is not satisfied. In this case, we use the original monthly LE values
without correction. A simplified flowchart of these steps is displayed in Fig. S3 in the supplementary
material.
In a further pre-processing step, if a site is located in close proximity to other sites such that they all sit
on the same 0.25° grid-cell, we use observational data from the site that is more representative of the
underlying grid-cell. Selecting the most representative site among these sites involves 1) identifying the
biome cover at each site; 2) computing the fraction of the grid area covered by each biome; the most
representative site is the one whose biome is more abundant in the underlying grid-cell (i.e. scores the
highest fraction of the total area). If all sites are equally representative of the underlying grid-cell, we
consider them as one site and we combine monthly LE from the sites by taking the average. We use the
high resolution 300 m - land cover maps from the European Space Agency (ESA; http://www.esa.int/)
downloaded from https://cds.climate.copernicus.eu/ to determine the biome types of neighbouring
sites and the corresponding grid-cells. This step has ensured that we are not matching a grid-cell with
inappropriate observational data. This filtering reduced the number of employed sites in this study from
366 sites to 260 sites (Fig. S1). All the excluded sites are in Europe and North America. Furthermore, we
exclude 6 sites from the weighting, located on flooded land area, wetlands or intensively irrigated land.
As a result of this, the constraining observational dataset used to derive DOLCE V2 includes 254 sites
with a total of 13641 monthly records.
## 2.2.   Methods
### 2.2.1 Weighting approach
The weighting technique is the same as that used in DOLCE V1 and was originally presented by Bishop and
Abramowitz (2013) and implemented for merging observational estimates by Hobeichi et al. (2018, 2019,
2020a). It consists of building a linear combination, $\mu$, of the parent datasets that minimise
$\sum_{j=1}^{J}(\mu^j - y^j)^2$, where $j \in [1, J]$ are the monthly time-site records, $y^j$ is the observed ET at the $j^{th}$ time-
site record. The linear combination $\mu^j = \sum_{k=1}^{K} w_k x_k^j$ is subject to the constraint that $\sum_{k=1}^{K} w_k = 1$,
where $k \in [1, K]$ represents the parent datasets and $x_k^j$ is the value of the $k^{th}$ bias-corrected parent
dataset (i.e. after subtracting its mean bias relative to the all-site observational dataset) corresponding to
the $j^{th}$ time-site record. The analytical solution to this problem accounts for both the performance
differences between the parent datasets and their error covariance, a proxy for dependence. Further
details on the merging technique can be found in Abramowitz and Bishop (2015) and Bishop and
Abramowitz (2013). The weighting approach is used to combine the global parent datasets separately on



different spatiotemporal subsets of the entire period and globe, using a tiered approach detailed in
section 2.2.3.
## 2.2.2 Computing uncertainty in ET
The ensemble dependence transformation process developed by Bishop and Abramowitz (2013) is used
to calculate the spatiotemporal uncertainty of DOLCE V2. The process transforms the global parent
datasets to a new ensemble so that the variance of the transformed ensemble about the derived hybrid
ET estimate, $\mu$, is constrained to be equal to the error variance of $\mu$ with respect to the flux tower data,
averaged over time and space (i.e. across all $J$ records). We use the spread $\sqrt{\sigma^2}$ of the transformed
ensemble as the spatially and temporally varying estimate of uncertainty standard deviation, which we
will refer to as uncertainty. We refer the reader to Bishop and Abramowitz (2013) for the derivation of
this approach, and Hobeichi et al (2018) for its implementation in this context. The spread $\sqrt{\sigma^2}$ of the
transformed ensemble accurately reflects the uncertainty of $\mu$ in those grid-cells where flux tower
observations are available. This process ensures that the computed uncertainty provides a better
uncertainty estimate of the hybrid ET than simply using the spread of the parent datasets.
One additional advantage of defining uncertainty in this way is that it should give an accurate upper
bound estimate of the likely discrepancy between the product and unseen ET measurements at a range
of spatial scales. That is, since it is based on the discrepancy of the final hybrid product and point-based
flux tower estimates, which are essentially at the extremes of spatial discrepancy, the discrepancy
between DOLCE and actual ET at any spatial scale greater than that of a tower footprint should be less
than this uncertainty estimate (noting however that this is the estimated standard deviation of
uncertainty, rather than a hard upper limit). In 2.2.6 below, we detail the out-of-sample testing of this
uncertainty estimate at the point scale.
## 2.2.3 Tiering of data set subsets in time and space to maximise coverage
To derive DOLCE V1 over the global land, we applied spatial tiering (using different subsets of parent
products in different regions to maximise spatial coverage). We now expand this approach to include
temporal tiering to improve the temporal reach of DOLCE. Collectively, the incorporated parent datasets
have a temporal cover over 1980 – 2018, but only a short common overlap during 2003-2007, and their
spatial intersection does not cover the global land. Therefore, to achieve a global land coverage from 1980
through 2018 without excluding any product, it was necessary to build DOLCE V2 from different subsets
of parent datasets in time periods and land regions depending on the availability of the parent datasets
as shown in Table 1. To this end, we consider 14 distinct temporal tiers. For example, tier 9 covers 2008 -
2012 and incorporates all datasets except SRB-GEWEX. Tier 1 incorporates the least parent datasets, for
the year 1980 (i.e. FLUXCOM-MET and GLEAM3.3A), while tier 8 uses all the parent datasets and covers
2003 – 2007. Furthermore, within each temporal tier, we consider three spatial sub-tiers, with each spatial
sub-tier covering a part of the land. These consist of (a) all land except Antarctica, Greenland and North
Africa, (b) only Antarctica and Greenland, (c) only North Africa. A similar spatial tiering approach was also
applied in DOLCE V1. Other spatial tiers, each consisting of a small number of grid cells were also
considered where necessary to ensure that no grid cell in DOLCE V2 is missing ET data if a single parent is



missing ET data for that grid cell. As a result of the tiering approach, weighting is computed separately
using a different subset of parent data sets and site data in each tier, resulting in distinct spatiotemporal
subsets of the entire period. Note that in the results we below, we briefly examine the extent to which
tiering results in temporal discontinuities. Collectively, the hybrid estimates developed throughout the
temporal tiers and their spatial sub-tiers form DOLCE V2 over the global land throughout 1980 – 2018.
### 2.2.4 Weighting groups
Previous studies have found that the performance of a global product can vary with different climatic
circumstances, suggesting that separating the weighting into separate regions or other groupings might
well improve the results of the weighting overall (Ershadi et al. 2014; Hobeichi et al. 2018; Michel et al.
2016). Grouped weighting simply involves dividing the time and/or space covered by a particular tier
into different subsets or groups (e.g with different climatic conditions), and then applying the weighting
technique separately for each group (within a single tier). We expect that grouped weighting has the
potential to improve weighting by accounting for the variation in performance of the parent datasets
over different climate or land conditions and can hopefully improve biases detected in DOLCE V1.
Hobeichi et al. (2018) tried to group flux tower sites based on their land cover type and computed
weights for each land cover type. However, this approach did not improve the results, whether grouping
by climate zone or aridity index, with the main reason being attributed to the small number of sites in
many groups. Despite the availability of 100 additional sites to constrain the weighting  here compared
to Hobeichi et al., (2018), the ratio of the observational data to the number of parents has not improved
across several climate or land cover types for this work. We therefore investigate new approaches to
grouped weighting that allow sufficiently low group numbers to keep a reasonable sample size in each
of them, including:
• Grouping by latitudinal zone: this is a simplification of grouping by climate type in which
climates are aggregated into three latitudinal zones: (i) high latitudes (±60° poleward),  (ii) mid-
latitudes ±60° towards the subtropics ±40°, and (iii) tropics and sub-tropics (between -40° and
40°). In each zone we apply a separate weighting using the corresponding group of sites.
• Grouping by continents: Sites are naturally separated by continental boundaries and we might
suspect that a particular ET product performs differently across continents. For instance,
precipitation is involved in the derivation of many of the parent datasets, and has been found
to have different fidelity over different continents (Hobeichi, Abramowitz, Contractor, et al.
350     2020).
• Grouping by hemisphere: Pan et al. (2020) found that ET estimates agree more in the Northern
hemisphere than in the Southern hemisphere. Therefore, performing separate weighting in each
hemisphere could be better than weighting across all global land.
• Grouping by seasons: Several studies have shown that the skill of ET datasets vary by seasons
(Jiménez et al. 2018; Long, Longuevergne, and Scanlon 2014; Mueller et al. 2011). To capture
these differences, we implement grouping by seasons and grouping by month (detailed below).
We consider two combined seasons i.e. summer-fall and winter-spring. In the summer-fall
season, we constrain the weighting with (1) monthly observations from sites located in the
Northern hemisphere during the period December–May, and (2) monthly observations from



sites located in the Southern hemispheres during the period June–November. The remaining
observational data is used to constrain the weighting during the winter-spring combined season.
•   Grouping by months: This is similar to grouping by seasons, the only difference is that the two
groups are June–November and December–May, without accounting for the different seasonal
phase between hemispheres.
### 2.2.5 Bias correction strategies
In DOLCE V1, we showed that part of the success of the weighting approach is due to the bias correction
applied before the weighting. Within each tier, the bias correction is applied simply by adding the mean
difference between a product and tower data uniformly to all values of a product before the weights are
derived – it is constant in space and time for a given product within one tier. The grouping strategies
detailed above examine the effect of considering different bias correction and weighting subgroups
within each spatiotemporal tier, with groups divided by region (continents or latitudes) or/and seasons.
As an alternative to the grouping strategies, we also investigate if deriving a spatially varying bias
correction within each tier could further improve the weighting. A spatially varying bias correction might
better capture the performance deficiencies of each the parent datasets.
To derive a global bias correction for a particular parent dataset within each tier, we first compute the
mean bias at each flux tower site across all the time records within the tier. We then assign those ET
bias values to the grid cells containing the sites. Finally, using the bias values at these grid cells, we
extrapolate the bias field to the entire global land domain within the tier using several different
extrapolation strategies, including inverse distance weighting (IDW), local polynomial interpolation and
nearest neighbourhood. As with the different weighting groups, we test the effectiveness of each
approach using out-of-sample tests, which we now describe.
### 2.2.6 Out-of-sample testing approach
To test the effectiveness of different weighting groups or bias-correction approaches, and assess which
strategy offers the best performance, we use out-of-sample tests. To do this, we first divide the flux
tower sites between the in-sample and out-of-sample groups by randomly selecting 25% of the sites as
out of sample. The remaining sites form the in-sample training set are used to compute bias correction
terms and weights for the parent datasets in each tier using the weighting technique without weighting
groups (as adopted in DOLCE V1), and with each of the groups and bias correction strategies detailed
above. In each case, these bias correction terms and weights are then applied to the parent datasets
and compared to the out-of-sample sites to test efficacy of the clustering or bias correction approach
employed. The process is repeated for each grouping or bias correction strategy to derive several hybrid
ET datasets for each out of sample group of sites.
For each strategy, the test was repeated 1000 times with a different random selection of sites being out
of sample. The performance of each hybrid ET estimate was evaluated across five statistical metrics.
These were root mean squared error (RMSE), absolute standard deviation difference $|\sigma_{dataset} -$





$\sigma_{observation}$|, correlation, mean absolute deviation (i.e. mean(|dataset – observation|)) and median
absolute deviation (median(|dataset – observation|)). DOLCE V1 has not been included in this test
because its coarser spatial resolution (i.e. 0.5°) excludes many coastal sites and so significantly reduces
the observational data we could use in this analysis. The out-of-sample test is carried out over the
common period of availability of all the parent datasets i.e. 2003 – 2007 to enable comparison of the
out-of-sample performance of each approach with all of the parent datasets.

We perform another out-of-sample experiment to test if the uncertainty estimate derived by the
successful grouping/bias correction strategy performs well out of sample. In this test, we first select a
site S, but instead of constraining the weighting using observed ET from this site, we compute the
weights and bias correction terms of the parent datasets by using all the sites except S (i.e. just one site
is out-of-sample). We then calculate the MSE of the derived hybrid ET against observations from all the
sites except S. We denote this value by uncertainty$_{in-sample}$, since it represents the uncertainty
estimate computed using the same observational dataset that we used to train the weighting. We also
calculate the MSE of the hybrid ET against the out-of-sample observations from S, and we denote this as
uncertainty$_{out-sample}$, since we perform the comparison against ET observations that have not been
used to train the weighting. We repeat this test for all the sites, and each time we calculate the ratio
$\frac{uncertainty_{in-sample}}{uncertainty_{out-sample}}$. In an ideal case, this ratio should equal to unity.


## 3. Results and Discussion


## 3.1 Selection of a grouping strategy

Figure 1 shows the out-of-sample performance of different grouping strategies (including no grouping)
against parent datasets (left column). The performance results across all 1000 different random site
samples are shown in a boxplot for each clustering method (yellow), non-clustered weighting (as per
DOLCE V1, in magenta, labelled NO.GROUPING), and each parent dataset (purple). The hybrid ET
estimates derived from grouping weighting are labelled LAT.ZONES, CONTINENTS, SEASONS, MONTHS,
and HEMISPHERE, following the grouping approaches outlined above. The plots in the left column show
that overall, the hybrid ET estimates outperform their parent datasets across all the performance
metrics and in all clustering settings. The hybrid ET estimates derived by implementing spatially varying
bias correction strategies failed to outperform the parent datasets in the out-of-sample site tests, and
have been excluded from the plot (Fig. S3). To highlight the differences between the grouping strategies,
we magnify the leftmost section of these plots in the right column of Fig. 1, and also show in red the
median value of each boxplot. Results only change slightly across the grouping approaches, with the
best results achieved by grouping weights by months. Despite the relatively small improvement offered
by this strategy at the out-of-sample sites over the other grouping strategies, we derive DOLCE V2
(Hobeichi 2020) by applying a grouped weighting by months. Recall that in this approach, the
observational and gridded ET data are split into two groups, one covering the period June – November





and the other covering December – May. Weighting and bias correction is then implemented in each
group separately for each tier to create the subsets from which the hybrid ET product is derived.
The box plots in Fig. 2 show the ratio $\frac{\text{Uncertainty}_{\text{in-sample}}}{\text{Uncertainty}_{\text{out-sample}}}$, obtained across the different grouping
techniques. Each boxplot represents this ratio from all sites out of sample and shows that over half of
the data, the ratio ranges between 0.83 and 1.51, with a median very close to 1. This confirms that,
overall, when the uncertainty estimates are computed out of sample, they are very similar to what they
would have been if they were computed in sample. Also, the fact the shift in ratio is mostly towards
values bigger than 1 rather than smaller than 1 indicates that $\text{Uncertainty}_{\text{in-sample}}$ is greater than
$\text{Uncertainty}_{\text{out-sample}}$ so that uncertainty is overestimated rather than underestimated. Interestingly,
the lower (0.86) and upper (1.46) quartiles achieved by grouping weighting by months are the closest to
1 than the other grouping techniques. This suggests that overall, grouping weighting by months is able
to derive slightly more robust uncertainty estimates than the other techniques.

## 3.2 Comparison of DOLCE V2 with its parent datasets

Figure 3 displays the latitudinal means of DOLCE V1, DOLCE V2 and its parent datasets computed over
land for 2003 – 2007. We exclude Antarctica from the analysis due to the lack of reliable reference
information on ET for validation. The grey ribbon in Fig. 3 represents the uncertainty of DOLCE V2
defined by the ± standard deviation interval. The pink shaded areas represent latitudinal domains where
some parent datasets do not estimate ET for some parts of the land. In the white area where all the
datasets have complete terrestrial coverage, the uncertainty standard deviation of DOLCE V2 mostly
contains the latitudinal variations of its parent datasets with the exception of FLUXCOM-RS which
exhibits larger ET over the tropics and subtropics of the southern hemisphere. This containment should
not be surprising since uncertainty estimates should be robust for point-scale estimates. Similarly, the
middle pink area shows that DOLCE V2 agrees with its parent datasets except BACI, FLUXCOM-RS,
FLUXCOM-MET and MOD16. These four datasets do not estimate ET over arid and semi-arid regions in
north Africa, the middle east and central Asia, so it is expected that they exhibit larger ET averages over
these latitudes. DOLCE V1 exhibits a slightly lower ET than DOLCE V2 in the tropics and sub-tropics.
DOLCE V2 appears in the lower end of the range of the other datasets from 60° poleward. All the
datasets exhibit considerable disparities over the mid-latitude south of -50°, where the contribution of
the terrestrial ET comes mostly from the lower Andes.
Figure 4 shows the spatial distribution of differences in the ET mean between DOLCE V2 and each of its
parent datasets. We apply different spatiotemporal masks for each comparison based on parent dataset
coverage (Table 1). We also compute the climatological difference of DOLCE V2 with its predecessor
DOLCE V1 over 2000 – 2009.
Over the temperate regions of the northern hemisphere, DOLCE V2 exhibits lower mean ET than all its
parents except SEBS. We have computed the mean bias of all these datasets relative to the
observational data available from sites located in these temperate latitudes. DOLCE V2 has a negligible





bias of 0.2 $W\ m^{-2}$ relative to the observational data. This bias results from a positive bias of 0.4 $W\ m^{-2}$
during June – November and a negative bias of -0.2 $W\ m^{-2}$ during December–May. All the parent
datasets except SEBS exhibit a positive bias that ranges between 2.7 and 11.4 $W\ m^{-2}$ and SEBS has a
negative mean bias of -3.4 $W\ m^{-2}$, that varies between -0.2 $Wm^{-2}$ during December – May and -6.3
$Wm^{-2}$ during June – November. We note that the bias relative to the in-situ observational datasets is
only indicative of the performance of the gridded datasets at the sites and do not necessarily represent
the actual mean bias over these regions. The discrepancy between DOLCE V2 and DOLCE V1 is relatively
small across all land.
Large differences between DOLCE V2 and FLUXCOM-RS are seen over the Congo and the Amazon basins,
southern Africa, and the Brazilian highlands. The mean climatological bias of FLUXCOM-RS relative to
observational data from these regions is 30 $W\ m^{-2}$. The bias is likely to be the reason for the
exceptionally large FLUXCOM-RS ET seen over the tropics in Fig. 3. On the other hand, DOLCE V2 exhibits
a much smaller bias than FLUXCOM-RS that ranges between 2.6 $W\ m^{-2}$ during June–November and 6.4
$W\ m^{-2}$ during December–May.
In general, there are apparent disparities in the patterns of climatological differences in the tropics
across all the maps. This results from the fact that global ET datasets exhibit large differences over the
tropics which has been highlighted previously (Paca et al. 2019; Pan et al. 2020), particularly over the
Amazon basin.

For reference, we provide in global maps of the seasonal climatology of DOLCE V2 computed throughout
1980 – 2018 in Fig. S5.

## 3.3 Comparison of basin and continental ET with existing literature
We now compare DOLCE V2 with annual mean ET aggregates over a range of river basins documented in
a recent study (Table 4 of Zhang et al., 2018). ET in this study - which we'll refer to as CDR-ET- is derived
by merging 10 available ET datasets into a hybrid ET which then receives corrections, so that the surface
water budget - established by derived hybrid estimates of the other hydrological variables - is closed.
Table 2 displays the mean annual ET aggregates in $mm\ year^{-1}$ across 20 river basins calculated for
DOLCE V2 and CDR-ET over the common period 1984 – 2010. Our results show that there is an overall
agreement across all the non-Siberian rivers where the difference in ET estimates is mostly around 10%.
The agreement worsens over the Arctic basins Indigirka, Kolyma, Lena, Northern Dvina, Yenisei and
particularly over Olenik and Pechora where the differences in ET estimates exceed 20%. Previous studies
have reported large uncertainties in the water fluxes over the Siberian basins (Lorenz et al. 2015) most
likely due to the absence of a proper representation of snow and permafrost dynamics (Candogan
Yossef et al. 2012). Interestingly, over the north American arctic basins Mackenzie and Yukon, DOLCE V2
and CDR-ET exhibit much smaller relative differences than at their Siberian counterparts.

We also compare DOLCE V2 with continental annual means of ET shown by L'Ecuyer et al. (2015). In
their study, they derive a hybrid ET by merging three global datasets. Then, they adjust the hybrid ET
and its associated uncertainty by enforcing the physical constraints of the surface and atmospheric
water and energy budgets using a data assimilation technique (DAT). Our results show that DOLCE V2
has smaller ET with larger associated uncertainties compared to those derived in L'Ecuyer et al. (2015)



(Table 3). The range of their ET estimate overlaps with the upper range of DOLCE V2 throughout all
continents. In L'Ecuyer et al. (2015), the uncertainty estimates are originally taken from the literature
and are deemed constant across time and space, then these are reduced by the DAT. The uncertainty
estimate of DOLCE V2 is firmly grounded in the spread of its parent ET datasets but is more robust than
this spread alone, since this spread has been recalibrated so that the uncertainty of DOLCE V2 relative to
the observational data is precisely the spread of the recalibrated parent datasets.
Finally, we compare DOLCE V2 with the ET component of Conserving Land Atmosphere Synthesis Suite
(CLASS; Hobeichi, 2019; Hobeichi et al., 2020a) which we denote as CLASS-ET. CLASS dataset comprises
coherent estimates of the surface water and energy budgets at the gridded monthly scale. CLASS-ET has
been derived by adjusting DOLCE V1 by enforcing the simultaneous closure of the surface water and
energy budgets using the same DAT as in  L'Ecuyer et al. (2015), and can be therefore considered an
improved version of DOLCE V1. Table S3 displays the continental area weighted averages of DOLCE V2,
DOLCE V1 and CLASS-ET and the mean differences DOLCE V2 – DOLCE V1 and DOLCE V2 – CLASS
computed over a common time period 2003-2009, and a using common spatial mask.  We find that, in
general, DOLCE-V2 is closer to CLASS-ET (i.e. the improved version of DOLCE V1), than DOLCE V1.
The average global land ET of DOLCE V2 during $1980 - 2018$   is $37\ W\ m^{-2}$. This falls in the lower range
of global ET climatology of $35 - 54\ Wm^{-2}$ computed across 20 ET datasets  during $1982 - 2011$   in Pan
et al. (2020)

## 3.4 Performance of DOLCE V2 at flux sites

We now compare DOLCE V2 with ET measured at the 260 sites used to derive it (Table S1). We display
two performance metrics - correlation and standard deviation - on a Taylor Diagram (Fig. 5). RMSE has
been excluded from the plot since the mean ET exhibited by DOLCE ET at a particular site does not
necessarily equal the mean observed ET at that site. All data has been normalised before computing the
statistical metrics so that the observational data at each site has a mean of zero and a standard
deviation of 1. Each coloured point summarises the performance statistics of DOLCE V2 at a single site.
The observational data is represented by a single "reference" point, i.e. the hollow point at one on the
horizontal axis. The plot in Fig. 5 shows that most of the coloured points lie close to the reference point,
indicating that DOLCE V2 is highly correlated with most of the observational data. Overall, Fig. 5 shows
good agreement with the observational datasets. Poor performance is seen over a small number of
sites. These are represented by points located outside the Taylor diagram area. Most of these sites have
less than one year of monthly records with several gaps, perhaps raising questions about observational
quality.
In further analysis, we investigate whether the performance of DOLCE V2 is reduced over a particular
land cover type. For this purpose, we repeat Fig. 5, but this time we colour-code the statistics points by
the land cover type of the sites they represent as shown in Fig. S6. The new plot does not reveal clear
links between the performance of DOLCE V2 and the biome types of the sites. Similarly, we could not
find performance links with the degree of representativeness of the site to the underlying grid-cell. This
is shown in Fig. S7 where colours represent the degree of agreement between the land cover type at the
footprint of the tower site and the dominant land cover of the grid-cell containing the site. As shown in
Fig. S7, we carry out this analysis on the basis of three levels of agreement. These include blue points


representing sites whose land types match the dominant land types of the underlying grid-cells; green
points representing sites whose land types cover more than 25% of the underlying grid-cells without
being the dominant land cover at these grid-cells; and pink points representing sites whose land types
covers less than 25% of the underlying grid-cells.
Finally, Fig. S8 shows timeseries of DOLCE V2 and observed ET at a selection of sites from various climate
types. Site properties are shown in Table S1.

## 3.5 Changes in ET since 1980

### 3.5.1 Annual ET trends over the global land

We produce a long-term (1980 – 2018) map of trends in annual ET totals (Fig. 6) as proposed by Mann-
Kendall (Kendall 1948; Mann 1945) using the Sen's slope method (Sen 1968). We use the uncertainty
estimates associated with the ET fields and the confidence interval of the slope as two confidence
measures to filter out spurious trends.
Unreliable trends occur in regions where ET uncertainty is high, such as in central Africa and Sahel, and
in the high latitudes where ET observations are sparse or do not exist. Inconsistent trend behaviour (CI
includes positive and negative values) is found in regions that experienced long phases of droughts and
non-droughts during 1980-2018, mainly in Australia, or a succession of drought and wet events, mainly
in southern United States and most of the Amazons basin (Marengo et al. 2018). As a result of this, a
general long trend in ET is not identified in these regions. Miralles et al. (2014) report that these changes
in ET over these regions reflect El-Niño-La-Niña cycle.  Similarly, we have not detected clear long trends
in southern South America and eastern and southern Africa. This partially agrees with the study of Pan
et al. (2020) where their figure 8 shows no  ET trend in eastern Africa, and no agreement on the sign of
trend between the participating datasets has been found in southern South America. Figure 6 indicates
that ET has intensified over most of the northern latitudes which has been highlighted in many studies
(e.g. Miralles et al. 2014; Pan et al. 2020; Zhang et al. 2016), and declined in western United States,
eastern India, most of Madagascar, and parts of the Ethiopian highland. Unfortunately, given the
absence of adequate in-situ observations that cover a long enough period to establish trends analysis, it
is difficult to validate the identified trends directly.
In further analysis, we examine whether the spatiotemporal tiering adopted in DOLCE V2 has resulted in
temporal discontinuities. Figure 7 illustrates the annual average line plot of the area weighted mean of
continental ET exhibited by DOLCE V2. The vertical dashed lines mark the beginning of a new tier (see
Table 1). While the line plot does shows some marked changes, we do not believe these reveal a signal
of temporal discontinuity, as most of the strong changes in ET that coincide with changes in tiers also
coincide with extreme events, and are specific to the continents where these events occurred. For
instance, the drop in ET in South America in 2016 is explained by an unprecedented drought over
tropical South America (Erfanian, Wang, and Fomenko 2017). Similarly, the drop in ET in Africa is caused
by several droughts occurring in many African regions since 2016. In Australia, the decline in ET since
2017 is caused by severe droughts that developed across most of Australia.



### 3.5.2 ET regimes


To understand changes in ET across wet and dry regions, we classify land into 6 distinct dry and wet ET
regimes according to two aspects of ET: annual averages and within-year relative variability. We apply K-
means clustering (MacQueen 1967) - an unsupervised machine learning algorithm known for its
outstanding efficiency in clustering data – by implementing the K-Means function and the least squares
quantisation method (Lloyd 1982) using R software.  K-Means identifies K centroids (i.e. imaginary
values representing the centre of the clusters) and assigns each data point to the cluster of the nearest
centroid using – in this paper - the least squares quantisation method. For each grid cell, we compute 1)
the average of the annual total ET across 39 years (1980-2018); 2) within-year relative variability
climatology by temporally averaging the relative standard deviation of monthly ET calculated over a year
and across all years. These have been used as input features for the unsupervised classification. After
trial and error, we find that the global land can be adequately classified into six distinct regimes that
include three dry and three wet regimes. According to centroids values (Table S4), we label the six
regimes from driest to wettest and we list the associated ET climatology and variability respectively: (i)
very low ET with high variability (3 mm, 14%), (ii) low ET with high variability (207mm, 10%) , (iii) mild
low ET with medium variability (371 mm, 7%), (iv) mild high ET with medium variability (631mm, 5%), (v)
high ET with low variability (913mm, 3%), and (vi) very high ET with low variability (1221, 1%). Figure 8
displays the spatial distribution of the 6 ET regimes.
We compare the derived ET regimes map with the modifed Köppen climate (KC) classification map by
Chen and Chen (2013). We find that each KC class overlaps with only one ET regime with only two
exceptions (Table 4): i) Land characterised by a 'Dry Steppe Hot arid' (coded BSh in KC)  climate belongs
the 'Mild low ET with medium variablity regime', but in two regions, the Indian Deccan plateau and
Argentinean Gran Chaco low forests, where the climate is BSh, the ET regime is 'Mild high ET with
medium variability'; ii) Regions with a 'Mild temperate Fully humid Hot summer' climate (coded Cfa in
KC) overlaps with the 'Mild high ET with medium variability' regime in coastal regions, and to the 'Very
high ET with low variability' regime in inland regions. These two KC classes (i.e. BSh and Cfa) are shown
in bold in Table 4. Overall, ET-regimes defined in this paper provide an efficient way to aggregate the KC
classes in less varied classes. This is not surprising knowing that KC classes are developed based on the
empirical relationship between climate and vegetation, and that ET links the water, energy (climate) and
carbon (vegetation) budgets.

### 3.5.3 Global annual trends across the ET regimes


We now explore annual trends in mean ET exhibited in each ET regime during 1980-2018. First, we
calculate the annual ET total climatology and ET relative variability climatology spatially averaged across
each regime separately, then we compute the trends in yearly ET as above (i.e. using Mann-Kendall and
the Sen's slope methods). Figure 9 illustrates trends' results for the dry regimes ( V.L.ET, H.variability,
L.ET, H.variability and M.L.ET, M.Variability) and the wet regimes (M.H.ET, M.variability, H.ET,
L.variability and V.H.ET, L.variability). Across all regimes, trends in yearly ET total are upward as
indicated by the positive signs of both the slopes and their complete confidence intervals. The strongest
trends occur in the 'M.L.ET, M.Variability' and the 'M.H.ET, M.variability' regimes at rates 1.8



$mm\ year^{-1}$ and 1.78 $mm\ year^{-1}$ respectively, while the slowest trend occurs in the 'V.L.ET
H.variability' regime where ET is in general low.
We repeat the same analysis for the 5 parent datasets that span at least 30 years. Sen's slope of the
trends and their confidence interval (computed at the 95% confidence level) are presented in Table 5.
Trends' behaviour is deemed inconclusive when the CI encompasses negative and positive values, in
which case trends are considered unreliable. These are presented with regular (as opposed to bold)
typeface and are exhibited by FLUXCOM-MET in all regimes. ERA5-land shows downward trends in the
'M.H.ET, M.variability' and 'H.ET, L.variability' regimes.  Both GLEAM 3.3A and PLSH show upward ET
trends in all regimes, with the exception of GLEAM which shows no reliable trends in the wettest ET
regime. Differences exist in the magnitude of trends as DOLCE V2 shows in general the strongest trends
across the majority of the regimes. As in DOCLE V2, the strongest trends in GLEAM 3.3A occur in the
'M.L.ET, M.Variability' and the 'M.H.ET, M.variability regimes'.
There are of course some notable limitations to the approach we have taken here, some of which were
previously discussed in Hobeichi et al. (2018). First, the weighting approach adopted here relies heavily
on flux tower observations, which can suffer from a range of technical issues (Burba and Anderson,
2010; Fratini et al., 2019), as well as temporal gaps during particular weather conditions such as
extremes (Van Der Horst et al. 2019), which can affect our results. Next, unresolved land surface
processes in the parent datasets due for example to the absence of a proper representation of snow and
permafrost dynamics, or the heterogeneity of the land surface are likely to lead to uncertain ET
estimation in DOLCE V2, since it is only a combination of its parent data sets. This applies particularly in
regions where observations are scarce or do not exist.

## 4. Conclusions

The derivation of a new hybrid ET dataset allowed us to examine historical trends in ET and their
robustness to observational uncertainty. The dataset, DOLCE V2, is publicly available and was the result
of several key improvements over its predecessor, incorporating more parent products, more in-situ
data, testing a range of alternative implementations of it weighting and bias correction approach,
increased spatial resolution and covers a longer time period. Despite the observationally constrained
approach to defining uncertainty, we found robust ET trends across most areas of the land surface,
enough to present a clear signal in each of the ET climate regimes we examined. These trends indicate a
global increase in land derived ET between 1980 and 2018. This contrasts with other gridded ET
products that did not incorporate the same degree of observational constraint in either their mean field
or uncertainty estimates, and demonstrates the usefulness of this long-term hybrid ET dataset.

## 5. Data Availability

DOLCE V2 dataset is available from the NCI data catalogue at
http://dx.doi.org/10.25914/5f1664837ef06; (Hobeichi 2020)



## 6. Competing interests.

The authors declare that they have no competing interests.

## 7. Acknowledgment

The authors acknowledge the support of the Australian Research Council Centre of Excellence for Climate Extremes (CE170100023). This research was undertaken with the assistance of resources and services from the National Computational Infrastructure (NCI), which is supported by the Australian Government. This work used eddy covariance data acquired and shared by the FLUXNET community, including these networks: AmeriFlux, AfriFlux, AsiaFlux, CarboAfrica, CarboEuropeIP, CarboItaly, CarboMont, ChinaFlux, Fluxnet-Canada, GreenGrass, ICOS, KoFlux, LBA, NECC, OzFlux-TERN, TCOS-Siberia, and USCCC. The FLUXNET eddy covariance data processing and harmonization was carried out by the ICOS Ecosystem Thematic Center, AmeriFlux Management Project and Fluxdata project of FLUXNET, with the support of CDIAC, and the OzFlux, ChinaFlux and AsiaFlux offices. Data were also obtained from the Atmospheric Radiation Measurement (ARM) Program sponsored by the U.S. Department of Energy, Office of Science, Office of Biological and Environmental Research, Climate and Environmental Sciences Division. This works used data sourced from Terrestrial Ecosystem Research Network (TERN) infrastructure, an Australian Government NCRIS enabled project; the Oak Ridge National Laboratory Distributed Active Archive Center (ORNL DAAC); the Land Cover project of the ESA Climate Change Initiative. We would like to thank all the principal investigators that authorised us to download site data from the European Fluxes Database, and all the research institutes that made publicly available and/or hosted the gridded ET datasets used in this study.

## 8. Tables

*Table 1: Spatial and temporal coverage and original resolution of the global ET datasets (at the time of analysis) used to develop DOLCE V2.1. The first column shows the number of temporal tier.*

| | Time period | BACI | ERA5-land | FLUXCOM-MET | FLUXCOM - RS | GLEAM 3.3A | GLEAM 3.3B | MOD16 | PML | PLSH | SEBS | SRB-GEWEX |
|---|---|---|---|---|---|---|---|---|---|---|---|---|
| Tier | Excluded Land domain | Antarctica Greenland North Africa | | Antarctica Greenland North Africa | Antarctica Greenland North Africa | | | Antarctica Greenland North Africa | Antarctica Greenland | | | |
| | Original resolution | 0.5° half hourly | 0.1° hourly | $\frac{1°}{12}$ monthly | $\frac{1°}{12}$ monthly | 0.25° monthly | 0.25° monthly | 0.05° monthly | 0.5° monthly | $\frac{1°}{12}$ monthly | 0.05° monthly | 0.1° 3-hourly |
| 1 | 1980 | | | ● | | ● | | | | | | |
| 2 | 1981 | | ● | ● | | ● | | | ● | | | |
| 3 | 1982 – 1983 | | ● | ● | | ● | | | ● | ● | | |





| | | | | | | | | | | | | |
|---|---|---|---|---|---|---|---|---|---|---|---|---|
| 4 | 1984 – 1999 | | • | • | | • | | | • | • | | • |
| 5 | 2000 1,2&3 | | • | • | | • | | • | • | • | | • |
| 6 | 2000 (4 – 12) | | • | • | | • | | • | • | • | • | • |
| 7 | 2001 – 2002 | • | • | • | • | • | | • | • | • | • | • |
| 8 | 2003 – 2007 | • | • | • | • | • | • | • | • | • | • | • |
| 9 | 2008 – 2012 | • | • | • | • | • | • | • | • | • | • | |
| 10 | 2013 | • | • | • | • | • | • | • | | • | • | |
| 11 | 2014 | • | • | • | • | • | • | • | | | • | |
| 12 | 2015 | | • | • | • | | • | • | | | • | |
| 13 | 2016 – 2017 (1 – 6) | | • | | | • | • | | | | • | |
| 14 | 2017 (7 – 12) – 2018 | | • | | | • | • | | | | | |

*Table 2: Mean annual ET aggregates in mm year$^{-1}$ across 20 river basins calculated for DOLCE V2 and CDR-ET over a common*
*period 1984 – 2010.*

| Basin | CDR-ET 1984 – 2010 | DOLCE V2 1984 – 2010 |
|---|---|---|
| Amazon | 1153 | 1167 |
| Amur | 295 | 309 |
| Columbia | 331 | 340 |
| Congo | 1045 | 1084 |
| Danube | 503 | 451 |
| Indigirka | 138 | 107 |
| Indus | 277 | 323 |
| Kolyma | 167 | 132 |
| Lena | 245 | 185 |
| Mackenzie | 241 | 214 |
| Mississippi | 577 | 513 |
| Murray-Darling | 411 | 419 |
| Niger | 401 | 456 |
| Northern Dvina | 324 | 232 |
| Ob | 323 | 245 |
| Olenek | 174 | 108 |
| Paraná | 892 | 854 |
| Pechora | 244 | 166 |





| | | |
|---|---|---|
| Yenisei | 265 | 216 |
| Yukon | 175 | 158 |


*Table 3: Annual continental averages of ET and its standard deviation uncertainty calculated for DOLCE V2 and developed in*
*(L'Ecuyer et al., 2015) over a common period 2000 – 2010.*

| continent | ET± uncertainty $(W\ m^{-2})$ (L'Ecuyer et al. 2015) $2000 - 2010$ | ET± uncertainty $(W\ m^{-2})$ DOLCE V2 $2000 - 2010$ |
|---|---|---|
| Africa | 45 ± 3 | 40 ± 17 |
| Australia | 27 ± 3 | 28 ± 16 |
| Eurasia | 33 ± 3 | 30 ± 13 |
| North America | 33 ± 6 | 28 ± 12 |
| South America | 77 ± 4 | 73 ±23 |


*Table 4: correspondence between ET-regimes derived here and Köppen climate classes derived in (Chen and Chen, 2013. Text in*
*bold fontface indicates that the Köppen climate is associated with more than one ET regime.*

| **ET regimes** | **Köppen climate classes** (Chen and Chen 2013) |
|---|---|
| Very low ET with high variability | Polar (Tundra/Frost) <br> Dry Desert (Hot/Cold) arid |
| Low ET with high variability | Snow Fully humid Cold summer/Cool summer <br> Snow Dry summer Cool summer <br> Snow Dry winter Cold summer <br> Dry Steppe Cold arid <br> Dry Desert Hot arid/Cold arid <br> Mild temperate Dry summer Cool summer <br> Mild temperate Dry summer Warm summer |
| Mild low ET with medium variability | Snow Fully humid (Hot/Warm summer) <br> Snow Dry winter (Hot/Warm/Summer) <br> **Dry Steppe Hot arid** <br> Mild temperate Dry summer Hot summer <br> Mild temperate Fully humid Warm summer |
| Mild high ET with medium variability | **Dry Steppe Hot arid (observed only in the Indian Deccan plateau and Argentinean Gran Chaco low forests)** <br> **Mild temperate Fully humid Hot summer (observed in inland regions)** <br> Mild temperate Dry winter (Hot/Warm summer) <br> Tropical Dry summer |
| High ET with low variability | **Mild temperate Fully humid Hot summer**/Warm summer (**observed in coastal regions**) <br> Tropical Dry winter |
| Very high ET with low variability | Tropical Fully humid <br> Topical Monsoon |

*Table 5: Trends in yearly ET total spatially averaged across each ET regime calculated for DOLCE V2 and its parents datasets that*
*have time-span of more than 30 years. The text shows slopes of the trend line and their confidence interval calculated at the*
*95% confidence level, bold text indicates that the confidence interval is strictly positive or negative.*




| Dataset and time span | V.L.ET, H.variability | L.ET, H.variability | M.L.ET, M.Variability | M.H.ET, M.variability | H.ET, L.variability | V.H.ET, L.variability |
|---|---|---|---|---|---|---|
| **DOLCE V2 1980-2018** | **0.3 [0.1, 0.5]** | **0.9 [0.5, 1.2]** | **1.8 [1.4, 2.1]** | **1.7 [0.9, 2.5]** | **1.3 [0.4, 2.3]** | **1.0 [0.3, 2.1]** |
| **ERA5-land 1981-2018** | -0.1 [-0.2, 0.01] | 0.04 [-0.2, 0.3] | -0.05 [-0.3, 0.2] | **-0.5 [-0.9, -0.2]** | **-0.8 [-1.1, -0.5]** | -0.08 [-0.3, 0.1] |
| **FLUXCOM-MET 1980-2014** | -0.01 [-0.07, 0.04] | 0.1 [-0.1, 0.3] | 0.1 [-0.05, 0.2] | 0.05 [-0.1, 0.2] | -0.04 [-0.2, 0.1] | -0.1 [-0.3, 0.1] |
| **GLEAM 3.3A 1980-2018** | **0.2 [0.1, 0.4]** | **0.7 [0.5, 1.0]** | **1.2 [1.0, 1.5]** | **1.3 [1, 1.6]** | **0.6 [0.4, 0.9]** | 0.3 [-0.1, 0.8] |
| **PML 1981-2012** | -0.1 [-0.3, 0.1] | **0.4 [0.1, 0.7]** | **1.0 [0.5, 1.4]** | 0.2 [-0.1, 0.6] | 0.08 [-0.4, 0.5] | -0.2 [-0.9, 0.6] |
| **PLSH 1982-2013** | **0.1 [0.03, 0.1]** | **0.4 [0.2, 0.6]** | **1.0 [0.6, 1.5]** | **1.4 [0.9, 1.9]** | **1.5 [0.9, 2.1]** | **0.9 [0.5, 1.5]** |



# 9. Figures
*Figure 1: Results of the out-of-sample test across five metrics of performance, (a) RMSE, (b) CORRELATION (c) SD difference , (d)*
*Mean Absolute Deviation , and (e) Median Absolute Deviation. Box plots represent spread over 1000 different selections of out-*
*of-sample sites. Different clustering methods (yellow) include:  no clustering (NO.GROUPING; shown in magenta and horizontal*
*dashed lines), by latitude (LATZONES), by continents (CONTINENTS), by seasons (SEASONS), by months (MONTHS), and by*
*hemisphere (HEMISPHERE), the red text marks the median values. Performance comparison with each of the parent datasets is*
*shown in purple.*





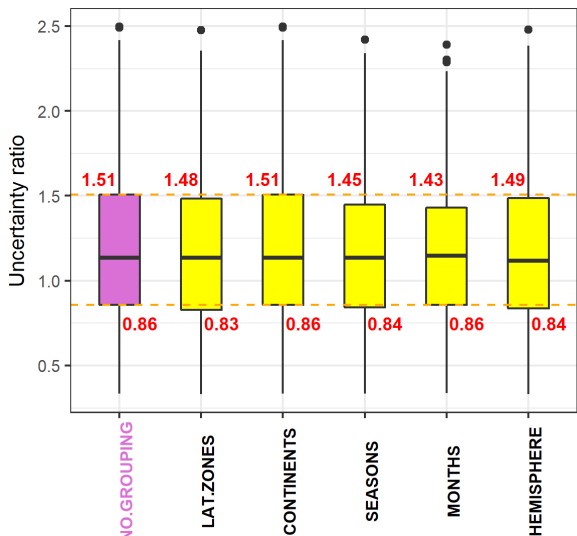


*Figure2: Box and whisker plots displaying the ratio* $\frac{Uncertainty_{in-sample}}{Uncertainty_{out-sample}}$, *computed for each site using the clustering methods*
*defined in Sect. 2.2.4. Labeling and colors are as in Fig. 1. Red text marks the value of the upper quantile (75%) and lower*
*quantile (25%).*



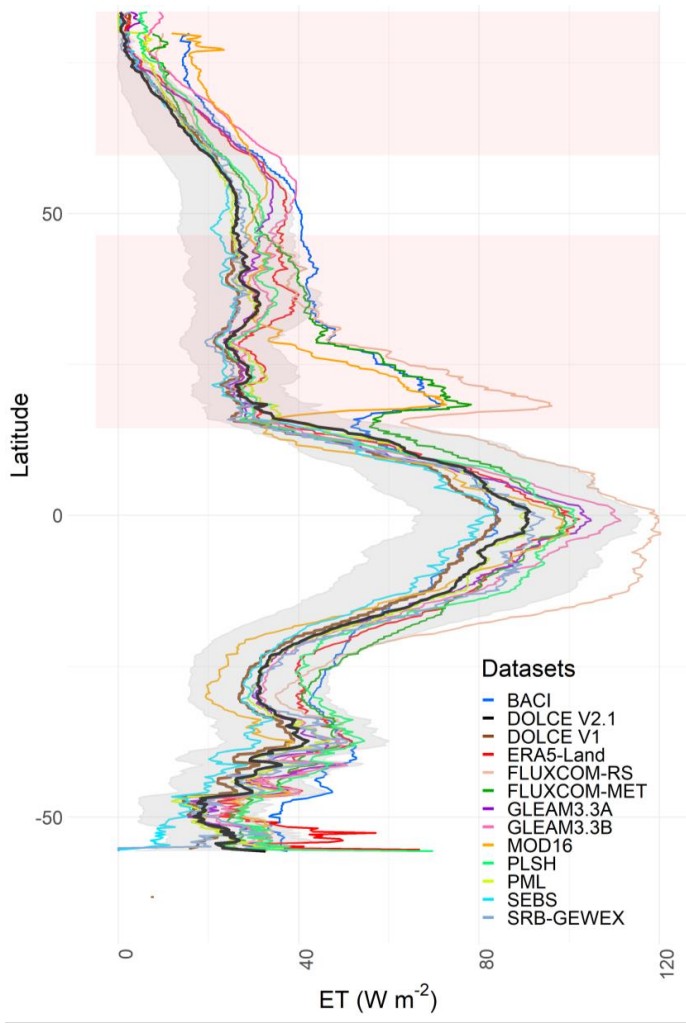

*Figure 3: Latitudinal means of DOLCE V2 and its parent datasets computed over a common period 2003–2007. The grey ribbon*
*represents the uncertainty standard deviation of DOLCE V2. The pink shaded areas represent latitudinal domains where some*
*parent datasets have gaps in some parts of the land as shown in Table 1.*



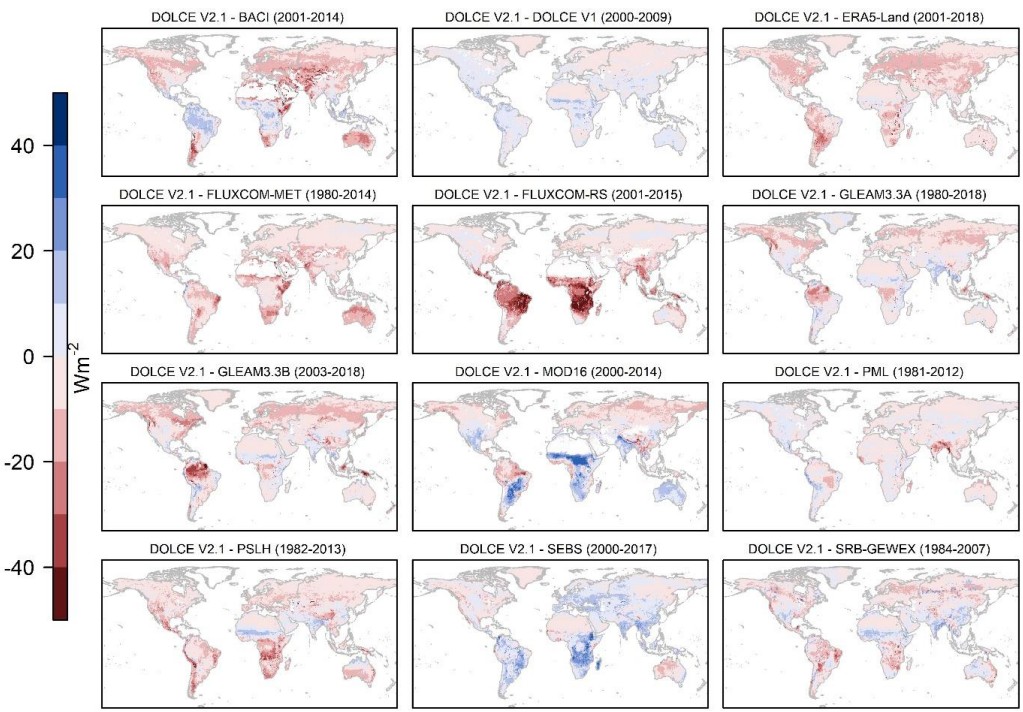

*Figure 4: Spatial distribution of differences in ET mean between DOLCE V2 and each of its parent datasets and DOLCE V1.*

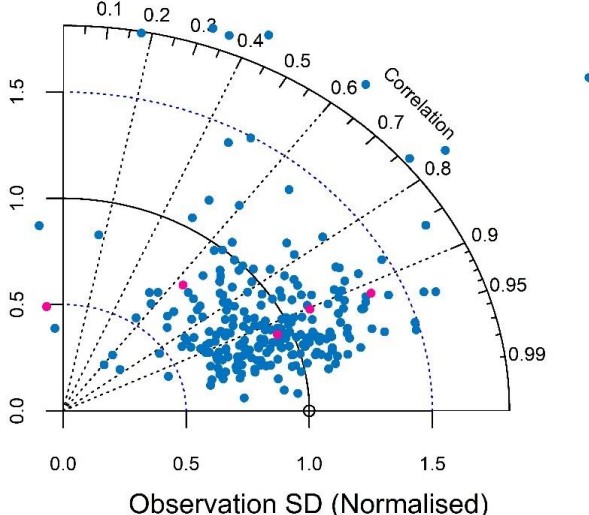

*Figure 5: Taylor Diagram displaying two performance metrics i.e. correlation and standard deviation of DOLCE V2 relative to*
*normalised observational data presented by a hollow point (reference point) at one unit on the x-axis. Pink points represent*
*performance statistics scored at sites located on wetlands, flooded plain or intensively irrigated areas.*





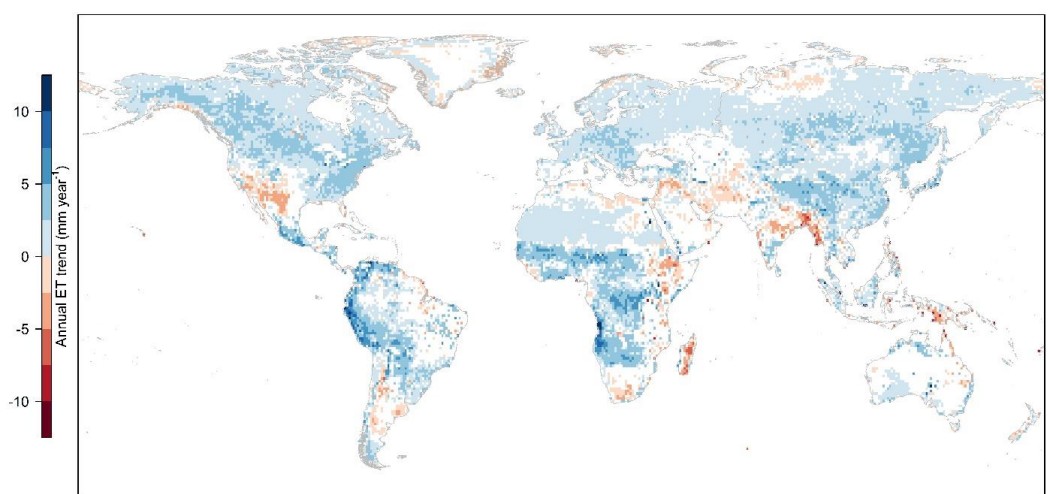

Figure 6: Spatial pattern of ET climate trends in DOLCE V2 over 1980 – 2018 as proposed by Mann-Kendall. Grid cells in white do not exhibit reliable trends as indicated by the implemented consistency measures.

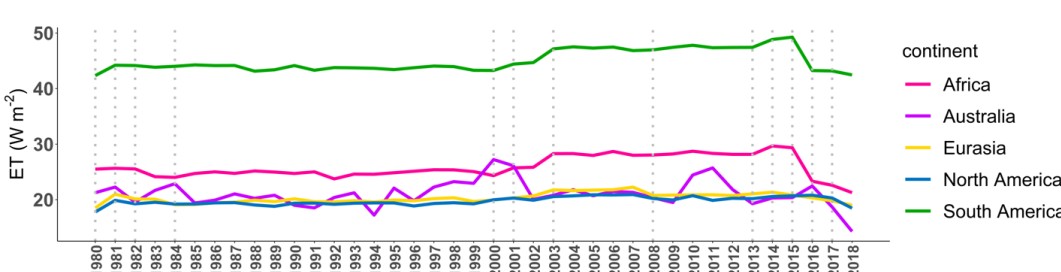

Figure 7: Annual average line plot for area weighted mean of continental ET.

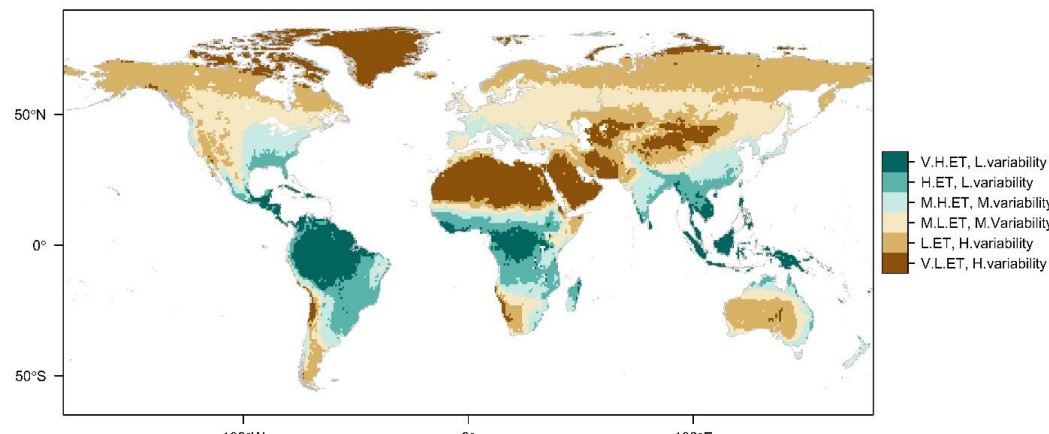

*Figure 8: Spatial distribution of ET regimes based on ET means and seasonal variations computed for 1980-2018.*

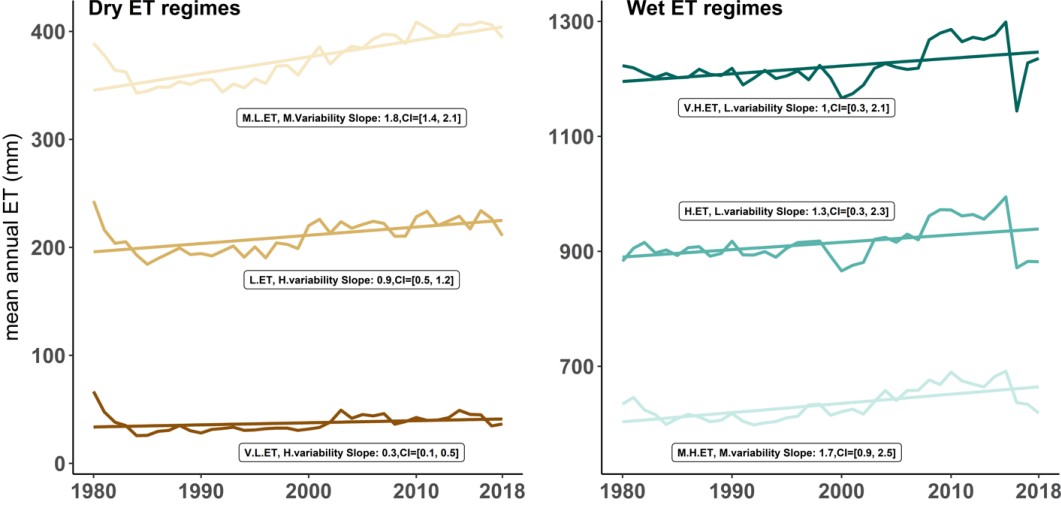

*Figure 9: Trends in mean global annual ET total computed for the dry and wet ET regimes during 1980-2018. Slopes and confidence intervals (CI) are computed at the 95% significance level.*



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
