# Peer review of "Robust historical evapotranspiration trends across climate regimes"

_Hydrology and Earth System Sciences, 2020_

## Referee Comment (RC1) · Jasper Denissen (Referee) · 22 Jan 2021

General comments

- The authors provide a new ET data set, making use of many parent ET data sets and in-situ ET measurements, at the same time avoiding information redundancy and exploiting information which is either missing in other parent data sets or has a large bias in comparison to the in-situ observations. The methodology provides a wealth of ways to improve DOLCE v2 as compared to DOLCE v1 and other ET data sets. They show both an impressive literature research and validation of this new ET data set. After the validation they compute historical trends, which show that across most of the regions globally, ET is to increase.

[Figure]

- Because of the efforts made to improve on DOLCE v1, a big part of the paper is about verifying DOLCE v2 against the parent datasets and in-situ observations. I think the title should reflect that.

- The authors put a lot of effort on trying to find ways to improve DOLCE v2 in comparison to DOLCE v1 by i) weighting groups (Figure 1; right column) and ii) Bias correction strategies (Figure 2; Supplementary Figure 3) as described in 2.2.4, 2.2.5 and 3.1. Despite the authors efforts to significantly improve the ET estimates, I think that the added complexity does not justify the little improvement gained. I would suggest moving sections 2.2.4, 2.2.5 and 3.1 and corresponding figures to the Supplementary materials. In the main text, the authors can shortly motivate the weighting group and bias correction strategy by referring to the Supplementary materials. This would make the derivation of DOLVE v2 ET more straight-forward and benefit the readability of the manuscript.

Specific comments

- Throughout the paper, please properly introduce i) table contents, ii) figure axes labels and color codes and iii) statistics of box-plots.

- lines 20: I found the notion that these climatology clusters / climate regimes are able to summarize or even replace the Köppen-Geiger climate regimes quite interesting. That would be worth a mention in the abstract, also putting this sentence into context.

- lines 23-25: "We find that despite robust . . . ET clusters". This is the only time this is mentioned in the entire manuscript. I don't see the relevance of it for the abstract.

- lines 82-84: FLUXCOM also belongs in this summation of gridded datasets that successfully exploit in-situ measurements.

- lines 198-200: I assume that the 'very large spatiotemporal domains' are equal to the spatial and temporal resolutions of the ET data sets? Do the authors mean that through time and varying wind directions, you might actually get closer to the grid cell mean than looking at individual days?

- lines 223-230: Just out of interest: What is the total amount of days initially available from all sites? After the filtering based on data availability, how many days are left?

- line 241: Could you elaborate on these conditions? As I'm not a flux tower measurements expert: Are these 20 and 30 W m-2 thresholds usually applied? Is there a paper where this methodology is also applied?

- lines 246-249: How do you justify using LE values without any correction – any value is better than nothing? Did you verify the differences between i) only LE data with correction and ii) LE data with and without correction? Are there any biases there?

- lines 250-255: Why not just take an average of the different towers within one grid cell, weighted by fractional cover of biome within that grid cell? I would assume that you want to have the possibility of retaining as much data as possible, as there could be data gaps in the data records in the tower measurements which could be filled with values from towers in the vicinity. And an additional question: Do you somehow account for varying footprints for the in-situ ET measurements? Depending on the location of the flux tower within the grid cell, what the tower measures could be from a neighboring grid cell.

- lines 272-275: Are w_k are the weights, which sum to 1, based on the error correlation in Supplementary Figure 2? If so, please state that here for clarity.

- lines 296-301: "the discrepancy between DOLCE and actual ET at any spatial scale greater than that of a tower footprint should be less than this uncertainty

estimate" I am slightly confused here: This would only be true for spatial scales greater than the tower footprint and smaller than 0.25x0.25 degree grid cell resolution (which is used in the study), right? If the spatial scale would be greater than 0.25x0.25 degrees, the discrepancy should be even larger?

- lines 351-353: This might be a naïve question, but is that not just because of the small number of flux towers on the SH? I could imagine that due to the higher availability of in-situ measurements and abundance of other measurements networks ET estimates are much more well constrained in the NH than the SH.

- lines 357-359: I was confused here, as I assumed the authors were referring to boreal summer. Please clarify.

- lines 380-282: I do not understand what is meant with 'extrapolating the bias field'. Please explain more clearly.

- Figure 3: Does zonal ET follow a Gaussian distribution? If not, it would make more sense to define the grey ribbon as the interquartile range instead of the standard deviation. Also, I would suggest making a mask of the grid cells where all of the parent data sets have values, so that a comparison is fairer. The figure without the mask could then be moved to the supplementary material.

- lines 500-501: Mentioning the seasonal cycle of DOLCE v2 ET but not elaborating on it feels out of place. Either elaborate on differences between seasonal cycles between DOLCE v2 ET and others or remove.

- lines 525-528: Please clarify this sentence; I found it confusing as written.

- lines 540-542: Either put into context by comparing with all the other literature references or remove. These two sentences seem lost.

- lines 546-550: In the first sentence the authors state the RMSE is not computed because the means between DOLCE and sites are not equal. In the sentence after you explain that all data has been normalized and therefore all have a zero mean. So, by normalizing you could in principle calculate the RMSE?

- lines 558-560: Would the signal of land cover be clearer when the authors would aggregate the land cover types to short/tall vegetation? Next to that, in Supplementary Figure 6, the color legend is blocking some of the extreme values.

- lines 571-572: Is there a specific conclusion drawn from these figures? Otherwise mentioning them seems unnecessary to me.

- Have the authors also looked at ET trends from flux tower measurements? As trends across all KS-clustering defined climate regimes are positive, the flux tower observations could corroborate that if they are also generally positive, right?

- line 678-680: I don't know how the fact that the global ET trends are different than the other ET products reflects usefulness; the fact that the DOLCE trends are different does not necessarily mean they are correct. However, it would be really interesting to see whether flux tower observations find similar long-term ET trends.

Technical corrections

- line 21: 'at each location'. Do you mean globally?

- line 42: remove the comma after 'approaches'

- line 113: replace 'trends (5) behavioural' with 'trends and (5) behavioural'

- lines 236: to avoid confusion, maybe rephrase "latent heat measurements are used directly" to "latent heat measurements are used without any corrections".

- line 435: Replace 'Fig. S3' with 'Fig S4.'

- line 590: replace 'intensified' with' increased'

- line 624: replace 'modifed' with 'modified'

- line 743: replace 'Figure2' with 'Figure 2'

I do not wish to remain anonymous - Jasper Denissen

---

## Referee Comment (RC2) · Anonymous Referee #2 · 2 Mar 2021

Review of "Robust historical evapotranspiration trends across climate regimes"

This paper presents a global estimate of ET including its uncertainties based on many sources of ET data. Compared to the previous version, this new version of "DOLCE" extends its coverage in time, resolution and inclusion of products. The new product is compared with earlier published ET estimates. In addition, the uncertainty estimates are evaluated. Also, historical trends in ET and their robustness is discussed. The latter results suggest a (robust) ET increases in recent years.

Overall this work seems like a useful addition to the literature. I have some detailed comments for clarification. The results and discussion section of the paper often for large parts mostly just list what is shown in the figures, but it would make it a lot more interesting to more read about what the figures teach us. In addition, please check if

small things table contents, figure axes, etc. are introduced. Often this seems somewhat lacking.

Detailed comments

L13: why "gridded"?

L19: "After successful evaluation of the efficacy": a "successful evaluation" does not say anything about the efficacy, so please rephrase.

L19: coverage, rather than reach?

L33: "with different scopes" is unclear in its meaning to me.

L35: "typically incorporating a range of remote sensing inputs" would benefit from some citations.

L36: "have been recognised for their potential to outperform single source datasets" can the strength of these methods be made in a more explicit statement that is more speficic?

L40: time resolution (rather than step)?

L43: chemical seems redundant?

L70: physically-based

L70: which ET trends did Pan look at?

L142-147: it seems some references could be added here?

Section 2.2.4. I do not suggest to redo the analysis, but why aren't weighing groups considered based on their physical similarity linked to ET (e.g. landcover) rather than these currently somewhat oddly chosen groups?

Table 1: indicate what a (lack of) marker means. It's somewhat obvious but it's still good to specify. . .

Table 3: why are uncertainties this large for DOLCE V2?

Table 5: specify unit of the trends.

Figure 1: is this necessary to include in the main paper, or could it be supplementary materials?

Figure 2: idem

Figure 3: can more distinguishable lines styles (i.e. color, thickness etc) be used better allow interpreting this figure?

L759: reliable or robust?

---

## Author Comment (AC1) · 28 Apr 2021

We would like to thank the referees for their constructive comments on our manuscript. This document outlines our responses to their comments. We provide a track changed version of the manuscript to highlight the changes made to the manuscript and the supplementary material. In addition to the suggested changes by the two referees, we have further improved the analysis by introducing a parallel, complementary dataset version to DOLCE V2.1, DOLCE V3, that has fewer parent datasets than V2.1, reducing the number of temporal tiers and temporal discontinuities found in DOLCE V2.1, mostly over the tropics. DOLCE V2.1 remains a more optimal dataset in many senses as it minimises bias and maximises correlation with in-situ

observation, whereas V3 prioritises temporal continuity. Similar to DOLCE V2.1, the superiority of DOLCE V3 over its parents is demonstrated using an out-of-sample testing approach. DOLCE V3 is presented alongside DOLCE V2.1 throughout the manuscript and has not resulted in any new sections or qualitative change to the manuscript. The main change is in section '3.5 Changes in ET since 1980', in which DOLCE V3 was used instead of DOLCE V2.1 to carry out the analysis of trends. The new results show that trends in DOLCE V3 ET are mostly within the range of trends in available ET datasets, unlike DOLCE V2.1 whose temporal inconsistencies resulted in higher trends than the available datasets mostly over the wet ET regimes. We have amended related text, figures and tables accordingly. These updated results also help to address the concerns of the referees, as outlined below.

================================================================================

Response to Referee 1, Jasper Denissen General Comments: ================= Because of the efforts made to improve on DOLCE v1, a big part of the paper is about verifying DOLCE v2 against the parent datasets and in-situ observations. I think the title should reflect that. We agree with the referee in that the technical side of the paper which includes improving DOLCE, comparing it with its parents, and verifying it against in-situ observations constitute a big part of the paper. However, given that we are not publishing this work as a data paper, we chose a title that highlights the scientific side of the this work, which is the robust analysis of trends in ET. Furthermore, most of the figures that show the performance of DOLCE against in-situ observations and highlight its superiority over its parent datasets have now moved to the supplementary material as suggested by both referees, and as a result of this, 4 out of the 7 figures included in the main manuscript are now focused on the assessment of trends. The authors put a lot of effort on trying to find ways to improve DOLCE v2 in comparison to DOLCE v1 by i) weighting groups (Figure 1; right column) and ii) Bias correction strategies (Figure 2; Supplementary Figure 3) as described in 2.2.4, 2.2.5 and 3.1. Despite the authors efforts to significantly improve the ET estimates, I think that the added complexity does not justify the little improvement gained. I would suggest moving sections 2.2.4,

2.2.5 and 3.1 and corresponding figures to the Supplementary materials. In the main text, the authors can shortly motivate the weighting group and bias correction strategy by referring to the Supplementary materials. This would make the derivation of DOLVE v2 ET more straight-forward and benefit the readability of the manuscript. We agree with the referee that the clustering methods added little improvement to the weighting, however a little improvement is better than nothing. Technically, grouped weighting requires aggregating the time and/or space domains prior to applying the weighting technique and adds a small amount of work and a few additional seconds of processing time. It is therefore worth investigating the efficacy of grouping approaches, and choosing the methods that adds the most improvement to the weighting even if the improvement is marginal. Furthermore, as detailed in the text, most of the weighting strategies have been previously suggested as ways to improve merging but have not been tested. Therefore, testing them here provides valuable information to the science community. To address the referee's concern and increase the readability of the manuscript, we have now moved sections '2.2.5 Bias correction strategies' and '3.1 Selection of a grouping strategy' and associated figures to the supplementary material.

===========================================================================
Specific comments: ================ Throughout the paper, please properly introduce i) table contents, ii) figure axes labels and color codes and iii) statistics of box-plots. We thank the referee for his comment. We have now explained the figures and the tables further throughout the text and in the captions. lines 20: I found the notion that these climatology clusters / climate regimes are able to summarize or even replace the Köppen-Geiger climate regimes quite interesting. That would be worth a mention in the abstract, also putting this sentence into context. We thank the referee for his suggestion, we have now mentioned the agreement of these classes with the Köppen-Geiger climate regimes in the abstract …. The new clusters include three wet and three dry regimes and provide an approximation of Köppen-Geiger climate classes. lines 23-25: "We find that despite robust . . . ET clusters". This is the only time this is mentioned in the entire manuscript. I don't see the relevance

of it for the abstract. Good point. We have now shown this finding in section 3.5.3 Global annual trends across the ET regimes . . . Our results indicate that decreasing ET trends observed in some regions oppose the consistent positive trends across the majority of ET clusters. lines 82-84: FLUXCOM also belongs in this summation of gridded datasets that successfully exploit in-situ measurements. Here we are listing the studies that applied fusion techniques attempting to match a global dataset that is deemed more reliable than the original datasets. FLUXCOM was listed earlier in the introduction with the machine-learning based datasets: . . .. techniques including machine-learning algorithms (Jung et al. 2010; Hamed Alemohammad et al. 2017; Jung et al. 2019), typically incorporating a range of remote sensing inputs. lines 198-200: I assume that the 'very large spatiotemporal domains' are equal to the spatial and temporal resolutions of the ET data sets? Do the authors mean that through time and varying wind directions, you might actually get closer to the grid cell mean than looking at individual days? We thank the referee for his comment. 'Very large spatiotemporal domains' means over many sites and time steps. This paragraph is trying to say that weights are derived by assessing the agreement between flux tower measurements and the value of underlying grid cells over many locations and time steps. This however assumes that the point scale of the flux towers can represent the grid scale. We don't expect this representativeness to be true at each site, however the ensemble of flux tower observations as a whole do represent the underlying grid cells. This has been thoroughly tested and validated in previous work. We have now made the paragraph clearer: First, weights for each product are constructed over very large spatiotemporal domains, i.e. more than 13000 space-time records as described below, so that the (assumed stochastic) biases of individual sites relative to grid cell values are unlikely to influence weights over a large sample. In fact representativeness of point-scale measurement for the grid scale does exist across all the flux tower sites as a whole, this has been verified by (Hobeichi et al. 2018). lines 223-230: Just out of interest: What is the total amount of days initially available from all sites? After the filtering based on data availability, how many days are left? The original sites data

was a mix of half-hourly, daily and monthly data. The majority of daily data come from Ameriflux sites. The raw Ameriflux data consisted of 147 sites with daily data and a total of 191,583 daily records. After quality control and filtering, the number of sites and daily records dropped to 56 and 81,142 respectively. line 241: Could you elaborate on these conditions? As I'm not a flux tower measurements expert: Are these 20 and 30 W m-2 thresholds usually applied? Is there a paper where this methodology is also applied? Good point, our response to this comment is now included in the text A study by Paca et al. (2019) examined the changes to flux tower LE by three means of corrections, and found that these on average differ by around 20 Wm-2 from one another. On this basis, we expect that typically, the correction of flux tower LE should not exceed 20 Wm-2, unless errors in other components of the budgets are propagating in the corrected ET. The rule for correcting small fluxes and the condition in which each rule is applied (i.e. LE= 30 Wm-2 ) are in part subjective and in another part based on a case by case assessment of changes induced to ET by the correction techniques, and achieve a reasonable trade-off between data quality and availability. lines 246-249: How do you justify using LE values without any correction – any value is better than nothing? Did you verify the differences between i) only LE data with correction and ii) LE data with and without correction? Are there any biases there? All ET measurements are prone to systematic errors, here we are using the physical constraints of the energy balance to minimize these errors. Constraining ET this way is a 'plus' rather than a 'must'. ET measurements, despite systematic errors in them, provide the most reliable information on ET and can still be used for ground-truthing gridded ET estimates. There is certainly a bias between i) and ii), however the majority of the sites, this bias is small relative to the values in gridded estimates. lines 250-255: Why not just take an average of the different towers within one grid cell, weighted by fractional cover of biome within that grid cell? I would assume that you want to have the possibility of retaining as much data as possible, as there could be data gaps in the data records in the tower measurements which could be filled with values from towers in the vicinity. And an additional question: Do you somehow account for

varying footprints for the in-situ ET measurements? Depending on the location of the flux tower within the grid cell, what the tower measures could be from a neighboring grid cell. Weighting ET measurements from two neighbouring towers located in two different biomes, with one dominant and another less dominant is expected to provide a better ET than only taking ET from the dominant biome. However, in all these cases, flux towers have different years of coverage. Therefore, considering data from both sites will retain more data but at the same time will lead to temporal inconsistency in the timeseries of the combined ET. No, we haven't accounted for varying footprints for the in-situ measurements. However, we have visually assessed the position of each flux tower in the grid cell using ArcGIS, and we have found that of the 260 sites used in this study, only two sites are close to the edge of the grid which means that their footprint might extend to that neighbouring grid cell. However, at both sites, the land cover and the climate within the expected footprint of the tower (< 2000 m) across the underlying and neighbouring grid cells is the same. lines 272-275: Are $w\_k$ are the weights, which sum to 1, based on the error correlation in Supplementary Figure 2? If so, please state that here for clarity. We thank the referee for his suggestion. We have now added a reference to Figure S2 in the text The analytical solution to this problem accounts for both the performance differences between the parent datasets and their error covariance (Fig. S2), a proxy for dependence. lines 296-301: "the discrepancy between DOLCE and actual ET at any spatial scale greater than that of a tower footprint should be less than this uncertainty estimate" I am slightly confused here: This would only be true for spatial scales greater than the tower footprint and smaller than 0.25x0.25 degree grid cell res- olution (which is used in the study), right? If the spatial scale would be greater than 0.25x0.25 degrees, the discrepancy should be even larger? The referee is right. We have now made this clear in the text . . . the discrepancy between DOLCE and actual ET at any spatial scale greater than that of a tower footprint and smaller than that of DOLCE should be less than this uncertainty estimate lines 351-353: This might be a naïve question, but is that not just because of the small number of flux towers on the SH? I could imagine that due to

the higher availability of in-situ measurements and abundance of other measurements net- works ET estimates are much more well constrained in the NH than the SH. This is true to a large degree for the data driven approaches such as machine learning ET estimates, which will have their reliability degraded in areas with less observations. However, we cannot assert that the lower availability of in-situ measurements in the Southern Hemisphere is driving disagreement among modelled and remote sensing ET estimates given that these typically do not rely on flux measurements. 357-359: I was confused here, as I assumed the authors were referring to boreal summer. Please clarify. We thank the referee for spotting this. Incorrect months were originally attributed to summer-fall and winter-spring seasons in the Northern and Southern hemispheres. We have now corrected the text by listing the correct months in each season We consider two combined seasons i.e. summer-fall and winter-spring. In the summer-fall season, we constrain the weighting with (1) monthly observations from sites located in the Northern hemisphere during the period June–November, and (2) monthly observations from sites located in the Southern hemispheres during the December–May. The remaining observational data is used to constrain the weighting during the winter-spring combined season. lines 380-282: I do not understand what is meant with 'extrapolating the bias field'. Please explain more clearly. The Bias field is the ET bias values described in lines 378 – 380. We have spatially extrapolated the ET bias values from the grid cells containing the sites to the entire global land. We have now made this clearer in the text. We then assign those ET bias values, or bias field, to the grid cells containing the sites. Finally, using the bias values at these grid cells, we extrapolate the bias field spatially to the entire global land domain within the tier using several different extrapolation strategies,... Figure 3: Does zonal ET follow a Gaussian distribution? If not, it would make more sense to define the grey ribbon as the interquartile range instead of the standard deviation. Also, I would suggest making a mask of the grid cells where all of the parent data sets have values, so that a comparison is fairer. The figure without the mask could then be moved to the supplementary material. We haven't examined the distribution of zonal ET,

and the uncertainty standard deviation term is not describing the latitudinal spread of ET around the mean (i.e. zonal standard deviation), but rather it shows the range of DOLCE ET values bounded by its uncertainty, i.e. DOLCE ± uncertainty. The uncertainty of DOLCE is explained in Section 2.2.2, and is described as the 'standard deviation of uncertainty'. We have now omitted 'standard deviation' from the caption to ensure that the reader does not misinterpret the grey ribbon and clarified in the text that the grey ribbon is 'DOLCE ± uncertainty'. We have also replaced this plot with a new version that applies the same spatial mask to all datasets. In the new figure, we have now added a similar plot for DOLCE V3 and its parents. lines 500-501: Mentioning the seasonal cycle of DOLCE v2 ET but not elaborating on it feels out of place. Either elaborate on differences between seasonal cycles between DOLCE v2 ET and others or remove. We thank the referee for his suggestions. We have now removed the plot of seasonal climatology lines 525-528: Please clarify this sentence; I found it confusing as written. We thank the referee for his comment. We have now clarified the sentence. The uncertainty estimate of DOLCE, however, is firmly grounded in the discrepancy between the gridded DOLCE product and in-situ tower data. The variance of this discrepancy is used to recalibrate the variance of the parent datasets, which are then used to estimate uncertainty, allowing spatiotemporally varying uncertainty estimate that is both consistent with the discrepancy between DOLCE and surface observations while at the same time being spatially and temporally complete. This process is detailed by Hobeichi et al (2018). lines 540-542: Either put into context by comparing with all the other literature references or remove. These two sentences seem lost. We thank the referee for his suggestion. We have now removed lines 540 - 542 from the text lines 546-550: In the first sentence the authors state the RMSE is not computed because the means between DOLCE and sites are not equal. In the sentence after you explain that all data has been normalized and therefore all have a zero mean. So, by normalizing you could in principle calculate the RMSE? We used the Taylor diagram to show how DOLCE performs at all sites, rather than a single site. To do this it was necessary to normalise the data because there will be different values of

the observed variable across different sites. Since the data is now normalised RMSE is not helpful here. To avoid any confusion, we have now removed this sentence from the paragraph. lines 558-560: Would the signal of land cover be clearer when the authors would aggregate the land cover types to short/tall vegetation? Next to that, in Supplementary Figure 6, the color legend is blocking some of the extreme values. We thank the referee for suggesting this. We have now combined 6 classes of broadleaved and needle leaved tree covers in one group, but we still could not find any clear signal. We have made the fill color of the legend transparent to make sure no values are covered. lines 571-572: Is there a specific conclusion drawn from these figures? Otherwise mentioning them seems unnecessary to me. We thank the referee for his suggestion. We have now removed these plots. Have the authors also looked at ET trends from flux tower measurements? As trends across all KS-clustering defined climate regimes are positive, the flux tower observations could corroborate that if they are also generally positive, right? Good point. Flux tower measurements have only started around 2002, and the longest observational records we had at any site was only 17 years. Therefore, there is no in-situ observations that cover a long enough period to examine trends in annual ET from observations. This was already acknowledged in the text in section 3.5.1 : Unfortunately, given the absence of adequate in-situ observations that cover a long enough period to establish trends analysis, it is difficult to validate the identified trends directly. line 678-680: I don't know how the fact that the global ET trends are different than the other ET products reflects usefulness; the fact that the DOLCE trends are different does not necessarily mean they are correct. However, it would be really interesting to see whether flux tower observations find similar long-term ET trends. We agree with the referee in that trends in DOLCE V2.1 are much stronger than those in other datasets. After some additional investigation we have made significant changes to this section, as explained below. However we do note that Table 5 shows that the global ET trends in the other datasets show differences in sign, magnitude and statistical significance of the trends across all ET regimes. We have now introduced DOLCE V3, a complementary dataset to

DOLCE V2.1, which we now use to carry out the analysis of trends instead of DOLCE V2.1. The main difference between the two datasets is the number of contributing parent datasets. The focus for DOLCE V3 was reducing the number of temporal tiers to reduce temporal discontinuities in DOLCE V2.1 (these were revealed in a separate analysis not related to this manuscript), mostly over the tropics. We believe these discontinuities and inhomogeneity lead to misrepresentative trends in some cases. We present V3 as a parallel version (rather than replacement) as V2.1 still remains a better performing data set in many of the out of sample tests. We have now repeated the analysis of trends in Section 3.5 and have shown new trends results incorporating DOLCE V3 (instead of DOLCE V2.1). The new results show that trends in DOLCE V3 agree with some products more than the others, and its trends' slopes are within the range of slopes of trends in the available products across all the ET regimes. This is now shown in the text and in the updated Table 5 We repeat the same analysis for all the participating parent datasets that span at least 30 years. Sen's slope of the trends and their confidence interval (computed at the 95% confidence level) are presented in Table 5. As noted earlier, trends' behaviour is deemed inconclusive when the CI encompasses negative and positive values. These are presented with regular (as opposed to bold) typeface and are exhibited by FLUXCOM-MET in all regimes except the driest. ERA5-land shows downward trends in the 'M.H.ET, M.variability' and 'H.ET, L.variability' regimes. Both GLEAM 3.5A and PLSH show upward ET trends in all regimes, with the exception of GLEAM which shows no reliable trends in the driest and wettest ET regimes. Differences exist in the magnitude of trends across the majority the products and the regimes. As in DOCLE V3, the strongest trends in GLEAM 3.5A occur in the 'M.H.ET, M.variability' regime at a rate 0.5 mm ãĂŰyearãĂŮ^(-1). Finally, the slopes of DOLCE V3 trends are within the range of slopes of trends in available ET products. Also, none of the available datasets incorporate the same degree of observational constraint in either their mean field or uncertainty estimates, which makes us believe that trends exhibited by DOLCE V3 are more reliable than those observed in other datasets. As we mentioned in our response to the previous

comment above, there are not enough in-situ observations that cover a long enough period to examine trends in annual ET directly from observations.

Technical corrections

line 21: 'at each location'. Do you mean globally?

Yes, we have now made it clear in the text by adding 'on land'. The sentence now reads:

. . . we derive novel ET climatology clusters for the land surface, based on the magnitude and variability of ET at each location on land.

line 42: remove the comma after 'approaches' We thank the referee for spotting this. We have now removed the coma after 'approaches'. line 113: replace 'trends (5) behavioural' with 'trends and (5) behavioural' Thank you for suggesting this. We have now made the change in the text. lines 236: to avoid confusion, maybe rephrase "latent heat measurements are used directly" to "latent heat measurements are used without any corrections". We thank the referee for his suggestion. We have now replaced 'directly' with 'without any corrections'. line 435: Replace 'Fig. S3' with 'Fig S4.' We thank the referee for spotting this. We have now made the change in the text. line 590: replace 'intensified' with' increased' We thank the referee for his suggestion. We have now made the change. line 624: replace 'modifed' with 'modified' We thank the referee for spotting this. We have now made the correction. line 743: replace 'Figure2' with 'Figure 2' We thank the referee for spotting this. We have now added a space after 'Figure'.

Please also note the supplement to this comment:
https://hess.copernicus.org/preprints/hess-2020-595/hess-2020-595-AC1-supplement.pdf

595, 2020.

a) 2003 - 2007

b) 2003 - 2016

**Datasets** (a)
- BACI
- DOLCE V2.1
- DOLCE V1
- ERA5-Land
- FLUXCOM-RS
- FLUXCOM-METa
- GLEAM3.3A
- GLEAM3.3B
- MOD16
- PLSH
- PML
- SEBS
- SRB-GEWEX

**Datasets** (b)
- DOLCE V2.1
- DOLCE V3
- ERA5-Land
- FLUXCOM-METb
- GLEAM V3.5A
- GLEAM V3.5B

Latitude

ET (W m$^{-2}$)

**Fig. 1.**

[Figure]

**Fig. 2.**

[Figure]

**Fig. 3.**

[Figure]

**Fig. 4.**

[Figure]

**Fig. 5.**

[Figure]

**Fig. 6.**

[Figure]

**Fig. 7.**

---

## Author Comment (AC2) · 28 Apr 2021

We would like to thank the referees for their constructive comments on our manuscript. This document outlines our responses to their comments. We provide a track changed version of the manuscript to highlight the changes made to the manuscript and the supplementary material. In addition to the suggested changes by the two referees, we have further improved the analysis by introducing a parallel, complementary dataset version to DOLCE V2.1, DOLCE V3, that has fewer parent datasets than V2.1, reducing the number of temporal tiers and temporal discontinuities found in DOLCE V2.1, mostly over the tropics. DOLCE V2.1 remains a more optimal dataset in many senses as it minimises bias and maximises correlation with in-situ observation, whereas V3

prioritises temporal continuity. Similar to DOLCE V2.1, the superiority of DOLCE V3 over its parents is demonstrated using an out-of-sample testing approach. DOLCE V3 is presented alongside DOLCE V2.1 throughout the manuscript and has not resulted in any new sections or qualitative change to the manuscript. The main change is in section '3.5 Changes in ET since 1980', in which DOLCE V3 was used instead of DOLCE V2.1 to carry out the analysis of trends. The new results show that trends in DOLCE V3 ET are mostly within the range of trends in available ET datasets, unlike DOLCE V2.1 whose temporal inconsistencies resulted in higher trends than the available datasets mostly over the wet ET regimes. We have amended related text, figures and tables accordingly. These updated results also help to address the concerns of the referees, as outlined below.

General Comments: Overall this work seems like a useful addition to the literature. I have some detailed comments for clarification. The results and discussion section of the paper often for large parts mostly just list what is shown in the figures, but it would make it a lot more interesting to more read about what the figures teach us. In addition, please check if small things table contents, figure axes, etc. are introduced. Often this seems somewhat lacking. We thank the referee for his comment. We have now explained the figures and the tables further throughout the text and in the captions. Detailed comments 1. L13: why "gridded"? The scale of ET observations from Eddy Covariance towers is typically less than 1000 m which does not allow to study ET at the regional scale. Therefore, gridded ET datasets are needed to understand ET at the regional scale. 2. L19: "After successful evaluation of the efficacy": a "successful evaluation" does not say anything about the efficacy, so please rephrase. We thank the referee for their suggestion. We have now changed "after successful evaluation of the efficacy of these uncertainty estimates out-of-sample" to: after demonstrating the efficacy of these uncertainty estimates out-of-sample

3. L19: coverage, rather than reach? We thank the referee for their suggestion. We have now changed 'reach' to 'coverage' 4. L33: "with different scopes" is unclear in its meaning to me. We thank the referee for their comment. We have now clarified 'with different scopes' in the text. …. with different scopes (e.g. addressing key questions in ecology, hydrology, or other disciplines), …. 5. L35: "typically incorporating a range of remote sensing inputs" would benefit from some citations. We thank the referee for their comment. Citation was already included before "typically incorporating a range of remote sensing inputs", we have now moved it to the end of the sentence. 6. L36: "have been recognised for their potential to outperform single source datasets" can the strength of these methods be made in a more explicit statement that is more speficic? We thank the referee for their suggestion. We have now specified the strength of merging methods. … have been recognised for their potential to outperform single source datasets in reducing bias against in reducing bias against tower-based eddy-covariance ET measurements

7. L40: time resolution (rather than step)? We thank the referee for their suggestion. We believe that both 'time resolution' and 'time step' can be used here interchangeably. 8. L43: chemical seems redundant? We thank the referee for their comment. However we couldn't find any redundancy. 9. L70: physically-based We thank the referee for spotting this, we have now made the correction in the text 10. L70: which ET trends did Pan look at? Pan looked at ET trends during 1982-2011. We have now specified this in the text. 11. L142-147: it seems some references could be added here? References of these products were given in the describing paragraphs that follow. We have now added those references in L142-147. 12. Section 2.2.4. I do not suggest to redo the analysis, but why aren't weighing groups considered based on their physical similarity linked to ET (e.g. landcover) rather than these currently somewhat oddly chosen groups? Good point. We agree with the referee in that the most obvious weighting groups are land cover and climate zones. We have tried both grouping methods in a paper describing the first version of DOLCE, but this did not improve the final hybrid product. This was explained in L336 – L337 Hobeichi et al. (2018) tried to group flux tower sites based on their land cover type and computed weights for each land cover type. However, this approach did not improve the results, whether grouping by climate zone or aridity index, with the main reason being attributed to the small number of sites in many groups. We disagree with the referee that the grouping approaches are 'somewhat oddly chosen'. We have clarified in 'Section 2.2.4 Weighting groups' the motivation behind each grouping method. We have now added a new weighting group that considers both physical similarities linked to ET, and seasons. We have applied this grouping to derive the new version 3 of DOLCE which we now use to examine ET trends. We have now explained the new grouping method in Section 2.2.4. • Grouping by ET regime and months: Land was classified into three distinct broad ET regimes (Fig. S4) according to two aspects of ET, mean annual total ET and within-year relative variability throughout 1980 – 2018, derived from GLEAM V3.5a, and using K-means unsupervised classification (MacQueen, 1967). We explain the classification method further in section 3.5.2. Different sets of weights were computed at each ET regime during June–November and December–May. Implementing weighting this way ensured that we account for performance differences across different physical aspects of the land and seasons. Despite that observational data was divided into six distinct groups, the observational data available in each group was still appropriate to merge the four parent datasets of DOLCE V3. However, we found this grouped weighting strategy not appropriate for merging 11 parent datasets of DOLCE V2. 13. Table 1: indicate what a (lack of) marker means. It's somewhat obvious but it's still good to specify. . . We are not sure what the referee is asking us to specify, but it seems from the referee's comment, the suggested change is not that important. . . .? 14. Table 3: why are uncertainties this large for DOLCE V2? DOLCE's uncertainty gives an accurate upper bound estimate of the likely discrepancy between the product and unseen ET measurements. Uncertainties in DOLCE V2 are large compared to uncertainties of hybrid estimates derived by different merging approaches which typically consider the spread the parent datasets. This has been clarified in Section 2.2.2

This process ensures that the computed uncertainty provides a better uncertainty estimate of the hybrid ET than simply using the spread of the parent datasets. One additional advantage of defining uncertainty in this way is that it should give an accurate upper bound estimate of the likely discrepancy between the product and unseen ET measurements at a range of spatial scales. That is, since it is based on the discrepancy of the final hybrid product and point-based flux tower estimates, which are essentially at the extremes of spatial discrepancy, the discrepancy between DOLCE and actual ET at any spatial scale greater than that of a tower footprint should be less than this uncertainty estimate (noting however that this is the estimated standard deviation of uncertainty, rather than a hard upper limit)

15. Table 5: specify unit of the trends. We thank the referee for their suggestion. We have now specified the unit of the trend as mm year-1. 16. Figure 1: is this necessary to include in the main paper, or could it be supplementary materials? We thank the referee for their suggestion. We have now moved this Figure to the supplementary material. 17. Figure 2: idem We thank the referee for their suggestion. We have now moved this Figure to the supplementary material. 18. Figure 3: can more distinguishable lines styles (i.e. color, thickness etc) be used better allow interpreting this figure? We thank the referee for their suggestions. We have now improved this figure and made the lines more distinguishable. 19. L759: reliable or robust? We have now rephrased the caption: Spatial pattern of ET climate trends in DOLCE V3 over 1980 – 2018 derived using Mann-Kendall and Sen's slope methods. Grid cells in white correspond to unreliable ET trends because (i) the confidence interval of the slope encompasses a mix of negative and positive values; or (ii) trends' slopes computed for multiple different random samples of ET within

Please also note the supplement to this comment:
https://hess.copernicus.org/preprints/hess-2020-595/hess-2020-595-AC2-supplement.pdf

———————————————

**Fig. 1.**

[Figure]

**Fig. 2.**

[Figure]

**Fig. 3.**

[Figure]

**Fig. 4.**

[Figure]

**Fig. 5.**

[Figure]

**Fig. 6.**

[Figure]

**Fig. 7.**

**Supplement:**

**Supplementary Material of:**

**Robust historical evapotranspiration trends across climate regimes**

Sanaa Hobeichi[1,2], Gab Abramowitz[1,2], and Jason Evans[1,2]

[1] Climate Change Research Centre, UNSW Sydney, NSW 2052, Australia.

[2] ARC Centre of Excellence for Climate Extremes, UNSW Sydney, NSW 2052, Australia.

Correspondence to: Sanaa Hobeichi (s.hobeichi@unsw.edu.au)

**S1.    Flux tower data**

Table S1: Observational data used to constrained the weighting and/or validate DOLCE V2. Site data was accessed during July – November, 2019. Data source includes Ameriflux (ameriflux.lbl.gov), the Atmospheric Radiation Measurement (ARM; arm.gov), AsiaFlux (asiaflux.net), European Fluxes Database (europe-fluxdata.eu), Fluxnet 2015, LaThuile Free Fair Use (fluxnet.fluxdata.org), Oak Ridge data repository (daac.ornl.gov), OzFlux (ozflux.org.au) and individual site principal investigators (PI).

| Site ID | Country | Longitude | Latitude | Data source | Reference |
|---|---|---|---|---|---|
| AR-SLu | Argentina | -66.4598 | -33.4648 | Fluxnet 2015 | (Cleverly et al., 2013) |
| AT-Neu | Austria | 11.3175 | 47.11667 | Fluxnet 2015 | (Cleverly et al., 2013) |
| AU-Ade | Australia | 131.1178 | -13.0769 | Fluxnet 2015 | (Beringer, 2013a) |
| AU-ASM | Australia | 133.249 | -22.283 | Fluxnet 2015 | (Cleverly et al., 2013) |
| AU-Cum | Australia | 150.7225 | -33.6133 | Fluxnet 2015 | (Pendall, 2015) |
| AU-CPr | Australia | 140.5891 | -34.0021 | OzFlux | (Calperum Tech, 2013) |
| AU-CTr | Australia | 145.4469 | -16.1032 | OzFlux | (Liddell, 2013a ) |
| AU-DaS | Australia | 131.388 | -14.1593 | Fluxnet 2015 | (Isaac, 2010) |
| AU-Dry | Australia | 132.3706 | -15.2588 | Fluxnet 2015 | (Beringer, 2013b) |
| AU-Emr | Australia | 148.4746 | -23.8587 | Fluxnet 2015 | (Schroder, 2014) |
| AU-Fog | Australia | 131.3072 | -12.5452 | Fluxnet 2015 | (Beringer, 2013c) |
| AU-How | Australia | 131.1523 | -12.4943 | OzFlux | (Andrykanus, 2012) |
| AU-Gin | Australia | 115.65 | -31.375 | Fluxnet 2015 | (Silberstein, 2015) |
| AU-GWW | Australia | 120.6542 | -30.1914 | OzFlux | (Macfarlane, 2013) |
| AU-Lox | Australia | 140.6551 | -34.4704 | Fluxnet 2015 | (Ewenz, 2015) |

| | | | | | |
|---|---|---|---|---|---|
| AU-RDF | Australia | 132.4776 | -14.5636 | Fluxnet 2015 | (Beringer, 2014a) |
| AU-Rig | Australia | 145.576 | -36.656 | Fluxnet 2015 | (Beringer, 2014a) |
| AU-Rob | Australia | 145.6302 | -17.1174 | Fluxnet 2015 | (Liddell, 2013b) |
| AU-Stp | Australia | 133.3503 | -17.1508 | Fluxnet 2015 | (Beringer, 2013d) |
| AU-TTE | Australia | 133.64 | -22.287 | Fluxnet 2015 | (Cleverly, 2013) |
| AU-Tum | Australia | 148.1516 | -35.6566 | Fluxnet 2015 | (Woodgate, 2013) |
| | | | | | |
| AU-Ync | Australia | 146.2907 | -34.9893 | OzFlux | (Beringer, 2013e) |
| AU-Wac | Australia | 145.1873 | -37.429 | Fluxnet 2015 | (Beringer, 2013f) |
| AU-Whr | Australia | 145.0294 | -36.6732 | OzFlux | (Beringer, 2017 ) |
| AU-Wom | Australia | 144.0944 | -37.4222 | Fluxnet 2015 | (Stefan, 2013) |
| BE-Bra | Belgium | 4.52056 | 51.30917 | Fluxnet 2015 | (Stoy et al., 2013) |
| BE-Lon | Belgium | 4.74613 | 50.55159 | Fluxnet 2015 | (McCaughey et al., 2006) |
| BE-Vie | Belgium | 5.99805 | 50.30507 | Fluxnet 2015 | (Stoy et al., 2013) |
| BNS | China | 101.2653 | 21.9275 | AsiaFlux | asiaflux.net |
| BR-Ban | Brazil | -50.1591 | -9.82442 | Oak Ridge | (Saleska et al., 2013) |
| BR-Ji1 | Brazil | -62.3572 | -10.7618 | Oak Ridge | (Saleska et al., 2013) |
| BR-Ji3 | Brazil | -61.9331 | -10.078 | Oak Ridge | (Saleska et al., 2013) |
| BR-Ma2 | Brazil | -60.2091 | -2.609 | Oak Ridge | (Saleska et al., 2013) |
| BR-Sa1 | Brazil | -54.959 | -2.857 | Oak Ridge | (Saleska et al., 2013) |
| BR-Sa3 | Brazil | -54.9714 | -3.01803 | LaThuile | (Stoy et al., 2013) |
| BW-Ma1 | Botswana | 23.56033 | -19.9165 | LaThuile | (Stoy et al., 2013) |
| CA-ARF | Canada | -83.955 | 52.7008 | AmeriFlux | (Euskirchen et al., 2017) |
| CA-Ca2 | Canada | -125.291 | 49.8705 | AmeriFlux | (Chen et al., 2006) |
| CA-Cbo | Canada | -79.9333 | 44.3167 | AmeriFlux | (Barr et al., 2002) |
| CA-Ca3 | Canada | -124.9 | 49.5346 | AmeriFlux | (Chen et al., 2006) |
| CA-Let | Canada | -112.94 | 49.7093 | AmeriFlux | (Conte et al., 2003) |
| CA-Man | Canada | -98.4808 | 55.8796 | LaThuile | fluxnet.ornl.gov |
| CA-Mer | Canada | -75.5186 | 45.4094 | LaThuile | (Lafleur, et al., 2003) |
| CA-NS2 | Canada | -98.5247 | 55.9058 | LaThuile | fluxnet.ornl.gov |
| CA-NS6 | Canada | -98.9644 | 55.9167 | LaThuile | fluxnet.ornl.gov |
| CA-NS7 | Canada | -99.9483 | 56.6358 | LaThuile | fluxnet.ornl.gov |
| CA-Ojp | Canada | -104.692 | 53.9163 | AmeriFlux | (Baldocchi et al., 2000) |
| CA-Qcu | Canada | -74.0365 | 49.2671 | LaThuile | (Chu et al., 2018) |
| CA-Qfo | Canada | -74.3421 | 49.6925 | Fluxnet 2015 | fluxnet.ornl.gov |
| CA-SCC | Canada | -121.299 | 61.3079 | AmeriFlux | (Euskirchen et al., 2017) |
| CA-SF1 | Canada | -105.818 | 54.485 | Fluxnet 2015 | fluxnet.ornl.gov |
| CA-SF3 | Canada | -106.005 | 54.0916 | Fluxnet 2015 | fluxnet.ornl.gov |
| CA-TP1 | Canada | -80.5595 | 42.6609 | AmeriFlux | (Arain et al., 2005) |
| CA-TP4 | Canada | -80.3574 | 42.7102 | AmeriFlux | (Arain et al., 2005) |
| CH-Cha | Switzerland | 8.41044 | 47.21022 | Fluxnet 2015 | fluxnet.ornl.gov |
| CH-Dav | Switzerland | 9.85592 | 46.81533 | Fluxnet 2015 | fluxnet.ornl.gov |

| | | | | | |
|---|---|---|---|---|---|
| CH-Fru | Switzerland | 8.53778 | 47.11583 | Fluxnet 2015 | fluxnet.ornl.gov |
| CH-Oe1 | Switzerland | 7.73194 | 47.28583 | LaThuile | fluxnet.ornl.gov |
| CN-Cng | China | 123.5092 | 44.5934 | Fluxnet 2015 | fluxnet.ornl.gov |
| CN-Du2 | China | 116.2836 | 42.0467 | Fluxnet 2015 | (Stoy et al., 2013) |
| CN-HaM | China | 101.18 | 37.37 | Fluxnet 2015 | (Li et al., 2013) |
| CN-QHB | China | 101.332 | 37.60743 | AsiaFlux | asiaflux.net |
| CN-QYZ | China | 115.0663 | 26.73396 | AsiaFlux | asiaflux.net |
| CZ-BK1 | Czech Republic | 18.53688 | 49.50208 | European Fluxes DB | europe-fluxdata.eu |
| CZ-BK2 | Czech Republic | 18.54285 | 49.49443 | European Fluxes DB | europe-fluxdata.eu |
| CZ-RAJ | Czech Republic | 16.69651 | 49.44372 | European Fluxes DB | europe-fluxdata.eu |
| CZ-Stn | Czech Republic | 17.9699 | 49.03598 | European Fluxes DB | europe-fluxdata.eu |
| CZ-wet | Czech Republic | 14.77035 | 49.02465 | Fluxnet 2015 | (Stoy et al., 2013) |
| DE-Geb | Germany | 10.9143 | 51.1001 | Fluxnet 2015 | (Revill et al., 2013) |
| DE-Hai | Germany | 10.453 | 51.07917 | Fluxnet 2015 | (Ershadi et al., 2014) |
| DE-Hte | Germany | 12.17611 | 54.21028 | European Fluxes DB | europe-fluxdata.eu |
| DE-Hzd | Germany | 13.48982 | 50.96403 | European Fluxes DB | europe-fluxdata.eu |
| DE-Kli | Germany | 13.52251 | 50.89288 | Fluxnet 2015 | (Revill et al., 2013) |
| DE-Lkb | Czech Republic | 13.30467 | 49.09962 | Fluxnet 2015 | fluxnet.ornl.gov |
| DE-Meh | Germany | 10.65547 | 51.27531 | LaThuile | fluxnet.ornl.gov |
| DE-RuR | Germany | 6.30413 | 50.62191 | Fluxnet 2015 | fluxnet.ornl.gov |
| DE-Seh | Germany | 6.44965 | 50.87062 | Fluxnet 2015 | fluxnet.ornl.gov |
| DE-SfN | Germany | 11.3275 | 47.80639 | Fluxnet 2015 | fluxnet.ornl.gov |
| DE-Wet | Germany | 11.45753 | 50.4535 | LaThuile | (Stoy et al., 2013) |
| DE-Zrk | Germany | 12.88901 | 53.87594 | Fluxnet 2015 | fluxnet.ornl.gov |
| DK-Fou | Denmark | 9.58722 | 56.4842 | LaThuile | (Stoy et al., 2013) |
| DK-Ris | Denmark | 12.09722 | 55.53028 | LaThuile | (Gilmanov et al., 2010) |
| DK-Sor | Denmark | 11.64464 | 55.48587 | Fluxnet 2015 | (Stoy et al., 2013) |
| DK-ZaF | Greenland | -20.5557 | 74.4791 | Fluxnet 2015 | (Soegaard and Nordstroem, 1999) |
| ES-ES1 | Spain | -0.31881 | 39.34597 | LaThuile | (Stoy et al., 2013) |
| ES-LgS | Spain | -2.96583 | 37.09794 | Fluxnet 2015 | fluxnet.ornl.gov |
| ES-LJu | Spain | -2.75212 | 36.92659 | Fluxnet 2015 | fluxnet.ornl.gov |
| ES-LMa | Spain | -5.77336 | 39.9415 | LaThuile | (Stoy et al., 2013) |
| ES-VDA | Spain | 1.4485 | 42.15218 | LaThuile | (Stoy et al., 2013) |
| FI-Hyy | Finland | 24.295 | 61.8475 | Fluxnet 2015 | (Stoy et al., 2013) |
| FI-Jok | Finland | 23.51345 | 60.8986 | Fluxnet 2015 | (Reichstein et al., 2005) |
| FI-Kaa | Finland | 27.29503 | 69.14069 | LaThuile | (Aurela et al., 2001) |
| FI-Kns | Finland | 24.35617 | 60.64683 | European Fluxes DB | europe-fluxdata.eu |
| FI-Lom | Finland | 24.20918 | 67.9972 | Fluxnet 2015 | fluxnet.ornl.gov |

| FI-Sod | Finland | 26.63783 | 67.36186 | Fluxnet 2015 | (Stoy et al., 2013) |
|--------|---------|----------|----------|--------------|---------------------|
| FI-Var | Finland | 29.61 | 67.7549 | European Fluxes DB | europe-fluxdata.eu |
| FR-Fon | France | 2.78014 | 48.4764 | LaThuile | (Stoy et al., 2013) |
| FR-Gri | France | 1.95191 | 48.84422 | Fluxnet 2015 | (Loubet et al., 2011) |
| FR-Hes | France | 7.06556 | 48.67416 | LaThuile | (Reichstein et al., 2005) |
| FR-LBr | France | -0.7693 | 44.71711 | Fluxnet 2015 | fluxnet.ornl.gov |
| FR-Lq1 | France | 2.73583 | 45.64306 | LaThuile | (Gilmanov et al., 2007) |
| FR-Lq2 | France | 2.73703 | 45.63919 | LaThuile | (Gilmanov et al., 2007) |
| FR-Lus | France | 0.12065 | 46.41425 | European Fluxes DB | europe-fluxdata.eu |
| FR-Pue | France | 3.59583 | 43.74139 | Fluxnet 2015 | (Wei et al., 2014) |
| GRW | Portugal | -28.0297 | 39.0911 | ARM | (ARM, 2009) |
| HFE | China | 116.782 | 32.5584 | ARM | (ARM, 2008) |
| HFK | S.Korea | 127.57 | 34.55389 | AsiaFlux | asiaflux.net |
| HU-He1 | Hungary | 16.65222 | 46.95583 | PI | (Barcza et al., 2009) |
| HU-Bug | Hungary | 19.6013 | 46.6911 | LaThuile | (Stoy et al., 2013) |
| HU-Mat | Hungary | 19.726 | 47.8469 | LaThuile | (Stoy et al., 2013) |
| ID-Pag | Malaysia | 114.036 | 2.345 | LaThuile | (Hirano et al., 2007) |
| IE-Ca1 | Ireland | -6.91814 | 52.85879 | LaThuile | (Stoy et al., 2013) |
| IE-Dri | Ireland | -8.75181 | 51.98669 | LaThuile | (Stoy et al., 2013) |
| IL-Yat | Israel | 35.0515 | 31.345 | LaThuile | (Reichstein et al., 2003) |
| IS-Gun | Iceland | -20.2167 | 63.8333 | LaThuile | fluxnet.ornl.gov |
| IT-BCi | Italy | 14.95744 | 40.52375 | Fluxnet 2015 | fluxnet.ornl.gov |
| IT-CA3 | Italy | 12.0222 | 42.38 | Fluxnet 2015 | fluxnet.ornl.gov |
| IT-Cas | Italy | 8.71752 | 45.07005 | LaThuile | fluxnet.ornl.gov |
| IT-Col | Italy | 13.58814 | 41.84936 | Fluxnet 2015 | (Stoy et al., 2013) |
| IT-Cpz | Italy | 12.37611 | 41.70525 | Fluxnet 2015 | (Wei et al., 2014) |
| IT-Isp | Italy | 8.63358 | 45.81264 | Fluxnet 2015 | fluxnet.ornl.gov |
| IT-Lav | Italy | 11.28132 | 45.9562 | Fluxnet 2015 | (Cescatti and Zorer, 2003) |
| IT-Lec | Italy | 11.26975 | 43.30359 | LaThuile | (Stoy et al., 2013) |
| IT-LMa | Italy | 7.58259 | 45.15258 | LaThuile | fluxnet.ornl.gov |
| IT-Mal | Italy | 11.70334 | 46.11402 | LaThuile | (Gilmanov et al., 2007) |
| IT-MBo | Italy | 11.04583 | 46.01468 | Fluxnet 2015 | (Gilmanov et al., 2007) |
| IT-Noe | Italy | 8.15146 | 40.60613 | Fluxnet 2015 | fluxnet.ornl.gov |
| IT-Non | Italy | 11.09109 | 44.69019 | LaThuile | fluxnet.ornl.gov |
| IT-PT1 | Italy | 9.06104 | 45.20087 | Fluxnet 2015 | (Stoy et al., 2013) |
| IT-Ren | Italy | 11.43369 | 46.58686 | Fluxnet 2015 | (Stoy et al., 2013) |
| IT-Ro3 | Italy | 11.91542 | 42.37539 | European Fluxes DB | europe-fluxdata.eu |
| IT-SRo | Italy | 10.28444 | 43.72786 | Fluxnet 2015 | fluxnet.ornl.gov |
| IT-Tor | Italy | 7.57806 | 45.84444 | Fluxnet 2015 | (Galvagno et al., 2013) |
| JP-TKY | Japan | 137.4231 | 36.14617 | AsiaFlux | (Hirata et al., 2008) |
| MAO | Brazil | -60.5981 | -3.21297 | ARM | (ARM, 2014) |

| | | | | | |
|---|---|---|---|---|---|
| MMF | Japan | 142.2613 | 44.3219 | AsiaFlux | asiaflux.net |
| MN-SKT | Mongolia | 108.6543 | 48.35186 | AsiaFlux | (Hirata et al., 2008) |
| MSE | Japan | 140.0269 | 36.054 | AsiaFlux | asiaflux.net |
| MX-Lpa | Mexico | -110.438 | 24.1293 | AmeriFlux | (Bell et al., 2012) |
| NIM | Niger | 2.1758 | 13.4773 | ARM | (ARM, 2005) |
| NL-Ca1 | Netherlands | 4.927 | 51.971 | LaThuile | fluxnet.ornl.gov |
| NL-Haa | Netherlands | 4.80556 | 52.00361 | LaThuile | fluxnet.ornl.gov |
| NL-Hor | Netherlands | 5.0713 | 52.24035 | Fluxnet 2015 | (Sulkava et al., 2011) |
| NL-Loo | Netherlands | 5.74356 | 52.16658 | Fluxnet 2015 | (Gash and Dolman, 2003) |
| NL-Mol | Netherlands | 4.63908 | 51.65 | LaThuile | (Gilmanov et al., 2007) |
| NO-Adv | Norway | 15.923 | 78.186 | Fluxnet 2015 | fluxnet.ornl.gov |
| NSA | USA | -156.608 | 71.325 | ARM | (ARM, 2011) |
| NZ-BFu | New Zealand | 171.9268 | -43.5918 | OzFlux | (Laubach, 2016) |
| PL-wet | Poland | 16.3094 | 52.7622 | LaThuile | (stoy et al., 2013) |
| PT-Esp | Portugal | -8.6018 | 38.6394 | LaThuile | (stoy et al., 2013) |
| PT-Mi1 | Portugal | -8.00006 | 38.54064 | LaThuile | (stoy et al., 2013) |
| PT-Mi2 | Portugal | -8.02455 | 38.4765 | LaThuile | (stoy et al., 2013) |
| RU-Cok | Russia | 147.4943 | 70.82914 | Fluxnet 2015 | (Stoy et al., 2013) |
| RU-Fyo | Russia | 32.92208 | 56.46153 | Fluxnet 2015 | (Stoy et al., 2013) |
| RU-Ha1 | Russia | 90.00215 | 54.72517 | Fluxnet 2015 | (Belelli Marchesini et al., 2007) |
| RU-Ha2 | Russia | 89.95664 | 54.77301 | LaThuile | (Belelli Marchesini et al., 2007) |
| RU-Ha3 | Russia | 89.07785 | 54.70455 | LaThuile | (Belelli Marchesini et al., 2007) |
| RU-Zot | Russia | 89.3508 | 60.8008 | LaThuile | (Eugster et al., 2000) |
| SD-Dem | Sudan | 30.4783 | 13.2829 | Fluxnet 2015 | fluxnet.ornl.gov |
| SE-Deg | Sweden | 19.55669 | 64.18197 | LaThuile | fluxnet.ornl.gov |
| SE-Faj | Sweden | 13.55351 | 56.26547 | LaThuile | (Stoy et al., 2013) |
| SE-Fla | Sweden | 19.45694 | 64.11278 | LaThuile | (Stoy et al., 2013) |
| SE-Nor | Sweden | 17.4795 | 60.0865 | LaThuile | (Stoy et al., 2013) |
| SE-Sk2 | Sweden | 17.84006 | 60.12967 | LaThuile | (Stoy et al., 2013) |
| SE-Svb | Sweden | 19.7745 | 64.25611 | European Fluxes DB | europe-fluxdata.eu |
| SGP | USA | -96.855 | 37.521 | ARM | (ARM, 1997) |
| SN-Dhr | Senegal | -15.4322 | 15.40278 | Fluxnet 2015 | fluxnet.ornl.gov |
| TWP | Australia | 130.881 | -12.486 | ARM | (ARM, 2013) |
| UK-EBu | UK | -3.20578 | 55.866 | LaThuile | (Stoy et al., 2013) |
| UK-ESa | UK | -2.85861 | 55.90694 | LaThuile | (Stoy et al., 2013) |
| UK-Gri | UK | -3.79806 | 56.60722 | LaThuile | (Stoy et al., 2013) |
| UK-Ham | UK | -0.8583 | 51.15353 | LaThuile | (Stoy et al., 2013) |
| UK-LBT | UK | -0.1389 | 51.5215 | European Fluxes DB | europe-fluxdata.eu |
| UK-PL3 | UK | -1.26667 | 51.45 | LaThuile | fluxnet.ornl.gov |
| UK-Tad | UK | -2.82864 | 51.2071 | LaThuile | fluxnet.ornl.gov |
| US-ADR | USA | -116.693 | 36.7653 | AmeriFlux | (Euskirchen et al., 2017) |
| US-AR1 | USA | -99.42 | 36.4267 | Fluxnet 2015 | fluxnet.ornl.gov |

| US-AR2 | USA | -99.5975 | 36.6358 | Fluxnet 2015 | fluxnet.ornl.gov |
|--------|-----|----------|---------|--------------|------------------|
| US-ARc | USA | -98.0401 | 35.54649 | Fluxnet 2015 | (Stoy et al., 2013) |
| US-ARM | USA | -97.4888 | 36.6058 | Fluxnet 2015 | (Bagley et al., 2017) |
| US-Aud | USA | -110.51 | 31.5907 | AmeriFlux | (Baldocchi et al., 2015) |
| US-Bar | USA | -71.2881 | 44.0646 | LaThuile | fluxnet.ornl.gov |
| US-Bkg | USA | -96.8362 | 44.3453 | AmeriFlux | (Euskirchen et al., 2017) |
| US-Blk | USA | -103.65 | 44.158 | AmeriFlux | (Euskirchen et al., 2017) |
| US-Blo | USA | -120.633 | 38.8952 | Fluxnet 2015 | (Reichstein et al., 2003) |
| US-Bo1 | USA | -88.2904 | 40.0062 | LaThuile | (Stoy et al., 2013) |
| US-Br3 | USA | -93.6936 | 41.9747 | AmeriFlux | (Chu et al., 2018) |
| US-Cop | USA | -109.39 | 38.09 | Fluxnet 2015 | fluxnet.ornl.gov |
| US-CPk | USA | -106.119 | 41.068 | AmeriFlux | (Chu et al., 2018) |
| US-Ctn | USA | -101.847 | 43.95 | AmeriFlux | (Euskirchen et al., 2017) |
| US-CZ2 | USA | -119.257 | 37.0311 | AmeriFlux | (Euskirchen et al., 2017) |
| US-Dix | USA | -74.4346 | 39.9712 | AmeriFlux | (Chu et al., 2018) |
| US-EML | USA | -149.254 | 63.8784 | AmeriFlux | (Belshe et al., 2012) |
| US-FPe | USA | -105.102 | 48.3077 | LaThuile | (Ershadi et al., 2014) |
| US-FR3 | USA | -97.99 | 29.94 | AmeriFlux | (Euskirchen et al., 2017) |
| US-Fuf | USA | -111.762 | 35.089 | AmeriFlux | (Amiro et al., 2010) |
| US-GLE | USA | -106.239 | 41.3644 | Fluxnet 2015 | fluxnet.ornl.gov |
| US-GMF | USA | -73.2333 | 41.9667 | AmeriFlux | (Chu et al., 2018) |
| US-Goo | USA | -89.8735 | 34.2547 | Fluxnet 2015 | fluxnet.ornl.gov |
| US-Ha1 | USA | -72.1715 | 42.5378 | LaThuile | (Barford et al., 2001) |
| US-Ho2 | USA | -68.747 | 45.2091 | LaThuile | (Wei et al., 2014) |
| US-IB2 | USA | -88.241 | 41.8406 | AmeriFlux | (Allison et al., 2005) |
| US-Ivo | USA | -155.75 | 68.4865 | AmeriFlux | (Euskirchen et al., 2017) |
| US-Kon | USA | -96.5603 | 39.0824 | AmeriFlux | (Antunes et al., 2001) |
| US-KS2 | USA | -80.6715 | 28.60858 | Fluxnet 2015 | fluxnet.ornl.gov |
| US-KUT | USA | -93.1863 | 44.995 | AmeriFlux | (Euskirchen et al., 2017) |
| US-Los | USA | -89.9792 | 46.0827 | Fluxnet 2015 | (Baker et al., 2003) |
| US-Me1 | USA | -121.5 | 44.5794 | Fluxnet 2015 | (Irvine & Hibbard, 2007) |
| US-Me2 | USA | -121.557 | 44.4523 | Fluxnet 2015 | (Irvine & Hibbard, 2007) |
| US-MMS | USA | -86.4131 | 39.3231 | Fluxnet 2015 | (Baldocchi et al., 2001) |
| US-MOz | USA | -92.2 | 38.7441 | LaThuile | (Stoy et al., 2013) |
| US-Mpj | USA | -106.238 | 34.4384 | AmeriFlux | (Anderson-Teixeira et al., 2011) |
| US-MRf | USA | -123.552 | 44.6465 | AmeriFlux | (Chu et al., 2018) |
| US-Myb | USA | -121.765 | 38.0498 | AmeriFlux | (Baldocchi et al., 2018) |
| US-Ne1 | USA | -96.4766 | 41.1651 | Fluxnet 2015 | fluxnet.ornl.gov |
| US-NR1 | USA | -105.546 | 40.0329 | Fluxnet 2015 | (Albert et al., 2017) |
| US-Oho | USA | -83.8438 | 41.5545 | Fluxnet 2015 | (Stoy et al., 2013) |
| US-PFa | USA | -90.2723 | 45.9459 | LaThuile | (Keppel et al., 2012) |
| US-Pon | USA | -97.1333 | 36.7667 | AmeriFlux | (Euskirchen et al., 2017) |
| US-Prr | USA | -147.488 | 65.1237 | Fluxnet 2015 | (Ikawa et al., 2015) |

| US-RC1 | USA | -117.078 | 46.7837 | AmeriFlux | (Chi et al, 2017a) |
|--------|-----|----------|---------|-----------|--------------------|
| US-RC3 | USA | -118.598 | 46.991 | AmeriFlux | (Chi et al, 2017a) |
| US-RC4 | USA | -116.949 | 46.758 | AmeriFlux | (Chi et al, 2017a) |
| US-RC5 | USA | -119.248 | 47.01 | AmeriFlux | (Chi et al, 2017b) |
| US-Rls | USA | -116.736 | 43.1439 | AmeriFlux | (Euskirchen et al., 2017) |
| US-Ro1 | USA | -93.0898 | 44.7143 | AmeriFlux | (Baker et al., 2003) |
| US-SCd | USA | -116.372 | 33.6518 | AmeriFlux | (Euskirchen et al., 2017) |
| US-SCf | USA | -116.772 | 33.8079 | AmeriFlux | (Euskirchen et al., 2017) |
| US-SCs | USA | -117.696 | 33.7343 | AmeriFlux | (Euskirchen et al., 2017) |
| US-Sdh | USA | -101.407 | 42.0693 | AmeriFlux | (Billesbach and Arkebauer, 2004) |
| US-Ses | USA | -106.744 | 34.3349 | AmeriFlux | (Anderson-Teixeira et al., 2011) |
| US-SFP | USA | -96.902 | 43.2408 | AmeriFlux | (Euskirchen et al., 2017) |
| US-Shd | USA | -96.6833 | 36.9333 | AmeriFlux | (Burba et al., 2001) |
| US-Skr | USA | -81.0776 | 25.3629 | AmeriFlux | (Barr et al., 2013) |
| US-Slt | USA | -74.596 | 39.9138 | AmeriFlux | (Chu et al., 2018) |
| US-SP1 | USA | -82.2188 | 29.7381 | AmeriFlux | (Burton et al., 2002) |
| US-SP2 | USA | -82.2448 | 29.7648 | LaThuile | (Castro et al., 2000) |
| US-SP3 | USA | -82.1633 | 29.7548 | AmeriFlux | (Castro et al., 2000) |
| US-SRM | USA | -110.866 | 31.8214 | AmeriFlux | (Barron-Gafford et al., 2013) |
| US-Srr | USA | -122.026 | 38.2006 | AmeriFlux | (Chu et al., 2018) |
| US-SuW | USA | -156.491 | 20.8246 | AmeriFlux | (Anderson et al., 2015) |
| US-Syv | USA | -89.3477 | 46.242 | Fluxnet 2015 | (Chu et al., 2018) |
| US-Twt | USA | -121.652 | 38.1055 | Fluxnet 2015 | (Baldocchi et al., 2018) |
| US-UMd | USA | -84.6975 | 45.5625 | AmeriFlux | (Chu et al., 2018) |
| US-Var | USA | -120.951 | 38.4133 | Fluxnet 2015 | (Baldocchi et al., 2004) |
| US-WBW | USA | -84.2874 | 35.9588 | LaThuile | (Baldocchi et al., 2004) |
| US-WCr | USA | -90.0799 | 45.8059 | Fluxnet 2015 | (Baker et al., 2003) |
| US-Wdn | USA | -106.262 | 40.7838 | AmeriFlux | (Euskirchen et al., 2017) |
| US-Whs | USA | -110.052 | 31.7438 | AmeriFlux | (Biederman, et al., 2016) |
| US-Wi4 | USA | -91.1663 | 46.7393 | LaThuile | (Hilton et al., 2014) |
| US-Wi6 | USA | -91.2982 | 46.6249 | Fluxnet 2015 | (Noormets et al., 2007) |
| US-Wjs | USA | -105.862 | 34.4255 | AmeriFlux | (Anderson-Teixeira et al., 2011) |
| US-Wkg | USA | -109.942 | 31.7365 | Fluxnet 2015 | (Biederman, et al., 2016) |
| US-Wrc | USA | -121.952 | 45.8205 | AmeriFlux | (Chu et al., 2018) |
| RU-YLF | Russia | 129.2414 | 62.255 | AsiaFlux | asiaflux.net |
| RU-YPF | Russia | 129.6506 | 62.24139 | AsiaFlux | asiaflux.net |
| ZA-Kru | South Africa | 31.4969 | -25.0197 | LaThuile | (King et al., 2003) |
| ZM-Mon | Zambia | 23.25278 | -15.4378 | Fluxnet 2015 | (King et al., 2003) |

[Figure]

Figure S1: Location of the 260 sites used to derive and validate DOLCE V2, color-coded by data source.
Data source includes Ameriflux (ameriflux.lbl.gov), the Atmospheric Radiation Measurement (ARM;
arm.gov), AsiaFlux (asiaflux.net), European Fluxes Database (europe-fluxdata.eu), Fluxnet 2015, LaThuile
Free Fair Use (fluxnet.fluxdata.org), Oak Ridge data repository (daac.ornl.gov), OzFlux (ozflux.org.au) and
individual site principal investigators (PI).

[Figure]

Figure S2: Error correlation between the participating parent datasets of DOLCE V2 when compared to
in-situ data from 260 sites. Large correlation (>0.5) between two datasets indicates that their errors are
highly dependent.

[Figure]

Figure S3: Flowchart illustrating the correction steps carried out for every monthly record of observed LE
at the flux tower sites to correct for energy balance non-closure.

**S2.        Global ET datasets**

Table S2: Access information and download data of the global ET datasets (also referred to as parent
datasets) used to develop DOLCE V2 and DOLCE V3.

| Dataset | Access to data | Download date |
|---|---|---|

| BACI | https://doi.org/10.17871/BACI.224 | 03-10-2019 |
|---|---|---|
| ERA5-land | https://doi.org/10.24381/cds.e2161bac | 05-06-2020 |
| FLUXCOM-RS | https://doi.org/10.17871/FLUXCOM_EnergyFluxes_v1 LE.RS.EBC-BWR.MLM-ANN.METEO-NONE.4320_2160.monthly | 02-10-2019 |
| FLUXCOM-MET or FLUXCOM-METa | https://doi.org/10.17871/FLUXCOM_EnergyFluxes_v1 LE.RS_METEO.EBC-BWR.MLM-MARS.METEO-GSWP3.720_360.monthly | 02-10-2019 |
| FLUXCOM-MET FLUXCOM-METb | https://doi.org/10.17871/FLUXCOM_EnergyFluxes_v1 LE.RS_METEO.EBC-ALL.MLM-ALL.METEO-CRUNCEP_v8.720_360.monthly | 15-03-2021 |
| GLEAM3.3A | www.GLEAM.eu | 02-09-2019 |
| GLEAM3.3B | www.GLEAM.eu | 02-09-2019 |
| GLEAM3.5A | www.GLEAM.eu | 18-03-2021 |
| GLEAM3.5B | www.GLEAM.eu | 18-03-2021 |
| MOD16 | http://files.ntsg.umt.edu/data/NTSG_Products/MOD16/ MOD16A2_MONTHLY.MERRA_GMAO_1kmALB/GEOTIFF_0.05degree/ | 01-10-2019 |
| PML | https://data.csiro.au/collections/#collection/CIcsiro:17375v2 | 09-04-2019 |
| PLSH | http://files.ntsg.umt.edu/data/ET_global_monthly/Global_8kmResolution/ | 30-09-2019 |
| SEBS | http://en.tpedatabase.cn/portal/MetaDataInfo.jsp?MetaDataId=249454 | 09-04-2019 |
| SRB-GEWEX | https://disc.gsfc.nasa.gov/datasets/WC_PM_ET_050_1/summary | 01-10-2019 |

**S3.  Weighting groups**

[revised manuscript text omitted]

## S6.     Comparison of DOLCE V3 with its parent datasets

[Figure]

Figure S9: Spatial distribution of differences in ET climatology between DOLCE V3 and each of its parent
datasets and DOLCE V2. Different spatiotemporal masks are applied for each comparison based on the
spatiotemporal coverage of DOLCE V3 and the other datasets.

# S7.      Performance of DOLCE V2 at flux sites

Figure S10: Taylor Diagram displaying two statistical metrics i.e., correlation and standard deviation of
DOLCE V2 relative to normalised observational data presented by a hollow point (reference point) at
one unit on the x-axis. Statistics points are color-coded by the land cover of the sites they represent.
Land covers at the site locations are based on land cover maps from the European Space Agency (ESA;

). All broadleaved and needleleaved tree covers were combined together in a single
land cover 'Tree cover broadleaved/needleleaved'.

[Figure]

Figure S11: Taylor Diagram displaying two statistical metrics i.e., correlation and standard deviation of
DOLCE V2 relative to normalised observational data presented by a hollow point (reference point) at
one unit on the x-axis. Blue points represent sites whose land types match the dominant land types of
the underlying grid-cells; green points represent sites whose land types cover more than 25% of the
underlying grid-cells without being the dominant land cover at these grid-cells; and pink points
representing sites whose land types covers less than 25% of the underlying grid-cells. Land cover types
at the sites' footprint and the underlying grid-cells are determined based on land cover maps from the
European Space Agency.

**155  S8.   Comparison of DOLCE V2 with DOLCE V1 and Conserving**
**156  Land Atmosphere Synthesis Suite (CLASS-ET)**

Table S3: Area weighted mean ET ($Wm^{-2}$) computed for DOLCE V2, DOLCE V1 and CLASS-ET and
averaged over each of Africa, Australia, Eurasia, North America, South America and the global land
excluding Antarctica. The differences between DOLCE V2 and each of DOLCE V1 and CLASS-ET are shown
in columns 5 and 6 respectively.

|  | DOLCE V2 | DOLCE V1 | CLASS | DOLCE V2 – DOLCE V1 | DOLCE V2 – CLASS |
|---|---|---|---|---|---|
| Africa | 40.1 | 35.8 | 36.3 | 4.3 | 3.8 |

| | | | | | |
|---|---|---|---|---|---|
| Australia | 25.4 | 23 | 26.1 | 2.4 | -0.7 |
| Eurasia | 29.3 | 28 | 27.7 | 1.3 | 1.6 |
| North America | 33.2 | 30.5 | 31.1 | 2.7 | 2.1 |
| South America | 73.3 | 68.3 | 71.2 | 5 | 2.1 |
| Global land excluding Antarctica | 38.4 | 35.7 | 36.3 | 2.7 | 2.1 |

**S9.    ET regimes**

Table S4: List of 6 ET regimes identified by unsupervised learning. Second column shows the
abbreviation given to each regime. Third and fourth columns display the statistics of the class's centroid:
Yearly ET total climatology (column 3) and relative within-year standard deviation of monthly ET
climatology (column 4).

| Regime description | Regime abbreviation | Yearly ET total climatology (mm) | Relative standard deviation (%) |
|---|---|---|---|
| Very low ET with high variability | V.L.ET, H.variability | 60 | 10.5 |
| Low ET with high variability | L.ET, H.variability | 250.8 | 8.7 |
| Mild low ET with medium variability | M.L.ET, M.variability | 435.4 | 7.2 |
| Mild high ET with medium variability | M.H.ET, M.variability | 680.3 | 4.7 |
| High ET with low variability | H.ET, L.variability | 973.7 | 2.7 |
| Very high ET with low variability | V.H.ET, L.variability | 1408.2 | 0.9 |

**S10.    References**

Albert, L. P., Keenan, T. F., Burns, S. P., Huxman, T. E., and Monson, R. K.: Climate Controls Over Ecosystem Metabolism:
Insights From A Fifteen-Year Inductive Artificial Neural Network Synthesis For A Subalpine Forest, Oecologia, 184(1), 25-41,
2017.
Allison, V. J., Miller, R. M., Jastrow, J. D., Matamala, R., Zak, D. R.: Changes In Soil Microbial Community Structure In A Tallgrass
Prairie Chronosequence, Soil Science Society Of America Journal, 69(5), 1412-1421, 2005.
Amiro, B. D., Barr, A. G., Barr, J. G., Black, T. A., Bracho, R., Brown, M., Chen, J., Clark, K. L., Davis, K. J., Desai, A. R., Dore, S.,
Engel, V., Fuentes, J. D., Goldstein, A. H., Goulden, M. L., Kolb, T. E., Lavigne, M. B., Law, B. E., Margolis, H. A., Martin, T.,

McCaughey, J. H., Misson, L., Montes-Helu, M., Noormets, A., Randerson, J. T., Starr, and G., Xiao, J.: (2010) Ecosystem Carbon
Dioxide Fluxes After Disturbance In Forests Of North America, J. Geophys. Res. Atmos., 115(G00K02).
Anderson, R. G., Tirado-Corbalá, R., Wang, D., and Ayars, J. E.: Long-Rotation Sugarcane In Hawaii Sustains High Carbon
Accumulation And Radiation Use Efficiency In 2nd Year Of Growth, Agriculture, Ecosystems & Environment, 199, 216-224,
2015.
Anderson-Teixeira, K. J., Delong, J. P., Fox, A. M., Brese, D. A., and Litvak, M. E.: Differential Responses Of Production And
Respiration To Temperature And Moisture Drive The Carbon Balance Across A Climatic Gradient In New Mexico, Glob. Chang.
Biol., 17(1), 410-424, 2011.
Andrykanus, R.: Howard Springs Understory_old_20131128 OzFlux: Australian and New Zealand Flux Research and Monitoring
hdl: 102.100.100/14224, 2012.
Antunes, M. A. H., Walter-Shea, E. A., and Mesarch, M. A.: Test Of An Extended Mathematical Approach To Calculate Maize Leaf
Area Index And Leaf Angle Distribution, Agric. For. Meteorol., 108(1), 45-53, 2001.
Arain, M. A. and Restrepo-Coupe, N.: Net Ecosystem Production In A Temperate Pine Plantation In Southeastern Canada, Agric.
For. Meteorol., 128(3-4), 223-241, 2005.
Bagley, J. E., Kueppers, L. M., Billesbach, D. P., Williams, I. N., Biraud, S. C., and Torn, M. S.: The Influence Of Land Cover On
Surface Energy Partitioning And Evaporative Fraction Regimes In The U.S. Southern Great Plains, J. Geophys. Res. Atmos.:
Atmospheres, 122(11), 5793-5807, 2017.
Atmospheric Radiation Measurement (ARM) user facility, updated hourly. Eddy Correlation Flux Measurement System
(30ECOR). 1997-04-25 to 2004-03-31, Southern Great Plains (SGP) Smileyberg, KS (ABLE) (A4). Compiled by R. Sullivan, M.
Pekour and E. Keeler. ARM Data Center. Data set accessed 2019-09-17 at doi:10.5439/1025039, 1997.
Atmospheric Radiation Measurement (ARM) user facility updated hourly. Quality Controlled Eddy Correlation Flux
Measurement (30QCECOR). 2005-11-26 to 2007-01-07, ARM Mobile Facility (NIM) Niamey, Niger (M1). Compiled by R. McCoy,
S. Xie and Y. Zhang. ARM Data Center. Data set accessed 2019-09-17 at doi:10.5439/1097546, 2005.
Atmospheric Radiation Measurement (ARM) user facility, updated hourly. Quality Controlled Eddy Correlation Flux
Measurement (30QCECOR). 2008-05-06 to 2008-12-28, ARM Mobile Facility (HFE) Shouxian, Anhui, China (M1). Compiled by R.
McCoy, S. Xie and Y. Zhang. ARM Data Center. Data set accessed 2019-09-17 at doi:10.5439/1097546, 2008.
Atmospheric Radiation Measurement (ARM) user facility, updated hourly. Quality Controlled Eddy Correlation Flux
Measurement (30QCECOR). 2009-04-15 to 2010-09-30, ARM Mobile Facility (GRW) Graciosa Island, Azores, Portugal; AMF1
(M1). Compiled by R. McCoy, S. Xie and Y. Zhang. ARM Data Center. Data set accessed 2019-09-17 at doi:10.5439/1097546,
2009.
Atmospheric Radiation Measurement (ARM) user facility, updated hourly. Quality Controlled Eddy Correlation Flux
Measurement (30QCECOR). 2011-09-16 to 2020-05-11, North Slope Alaska (NSA) Barrow, Alaska (71.325, -156.608, 5) (E10).
Compiled by R. McCoy, S. Xie and Y. Zhang. ARM Data Center. Data set accessed 2019-09-17 at doi:10.5439/1097546, 2011.
Atmospheric Radiation Measurement (ARM) user facility, updated hourly. Eddy Correlation Flux Measurement System
(30ECOR). 2013-12-05 to 2015-01-10, Tropical Western Pacific (TWP) East Arm, Darwin, Australia (E30). Compiled by R. Sullivan,
D. Cook and E. Keeler. ARM Data Center. Data set accessed 2019-09-17 at doi:10.5439/1025039, 2013.
Aurela, M., Laurila, T. and Tuovinen, J.-P.: Seasonal CO2 balances of a subarctic mire, J. Geophys. Res. Atmos., 106(D2), 1623–
1637, doi:10.1029/2000JD900481, 2001.
Baker, I., Denning, A. S., Hanan, N., Prihodko, L., Uliasz, M., Vidale, P., Davis, and K., Bakwin, P.: Simulated And Observed Fluxes
Of Sensible And Latent Heat And CO2 At The WLEF-TV Tower Using SiB2.5, Glob. Chang. Biol., 9(9), 1262-1277, 2003.
Baldocchi, D. D., Law, B. E., and Anthoni, P. M.: On Measuring And Modeling Energy Fluxes Above The Floor Of A Homogeneous
And Heterogeneous Conifer Forest, Agric. For. Meteorol., 102(2-3), 187-206, 2000.
Baldocchi, D., Falge, E., Gu, L., Olson, R., Hollinger, D., Running, S., Anthoni, P., Bernhofer, C., Davis, K., Evans, R., Fuentes, J.,
Goldstein, A., Katul, G., Law, B., Lee, X., Malhi, Y., Meyers, T., Munger, W., Oechel, W., Paw, K. T., Pilegaard, K., Schmid, H. P.,
Valentini, R., Verma, S., Vesala, T., Wilson, K. and Wofsy, S.: FLUXNET: A New Tool to Study the Temporal and Spatial Variability
of Ecosystem–Scale Carbon Dioxide, Water Vapor, and Energy Flux Densities, Bull. Am. Meteorol. Soc., 82(11), 2415–2434,
doi:10.1175/1520-0477(2001)082<2415:FANTTS>2.3.CO;2, 2001.
Baldocchi, D. D., Xu, L., and Kiang, N.: How Plant Functional-Type, Weather, Seasonal Drought, And Soil Physical Properties Alter
Water And Energy Fluxes Of An Oak–Grass Savanna And An Annual Grassland, Agric. For. Meteorol., 123(1-2), 13-39, 2004.
Baldocchi, D. and Sturtevant, C.: Does day and night sampling reduce spurious correlation between canopy photosynthesis and
ecosystem respiration?, Agric. For. Meteorol., 207(), 117-126, 2015.
Baldocchi, D., Penuelas, J.: The Physics And Ecology Of Mining Carbon Dioxide From The Atmosphere By Ecosystems, Glob.
Chang. Biol., 256-257(2), 179-195, 2018.
Barcza, Z., Kern, A., Haszpra, L. and Kljun, N.: Spatial representativeness of tall tower eddy covariance measurements using remote sensing and footprint analysis, Agric. For. Meteorol., 149(5), 795–807, doi:10.1016/j.agrformet.2008.10.021, 2009.

Barford, C. C., Wofsy, S. C., Goulden, M. L., Munger, J. W., Pyle, E. H., Urbanski, S. P., Hutyra, L., Saleska, S. R., Fitzjarrald, D., and
Moore, K.: Factors Controlling Long- And Short-Term Sequestration Of Atmospheric CO2 In A Mid-Latitude Forest, Science,
294(5547), 1688-1691, 2001.

Barr, A. G., Griffis, T. J., Black, T. A., Lee, X., Staebler, R. M., Fuentes, J. D., Chen, Z., Morgenstern, K.: Comparing The Carbon
Budgets Of Boreal And Temperate Deciduous Forest Stands, Canadian Journal Of Forest Research, 32(5), 813-822, 2002.

Barr, J. G., Engel, V., Fuentes, J. D., Fuller, D. O., Kwon, H.: Modeling Light Use Efficiency In A Subtropical Mangrove Forest
Equipped With Co2 Eddy Covariance, Biogesciences, 10(3), 2145-2158, 2013.

Barron-Gafford, G. A., Scott, R. L., Jenerette, G. D., Hamerlynck, E. P., Huxman, T. E. (2013) Landscape And Environmental
Controls Over Leaf And Ecosystem Carbon Dioxide Fluxes Under Woody Plant Expansion, Journal Of Ecology, 101(6), 1471-1483

Belshe, E. F., Schuur, E. A., Bolker, B. M., and Bracho, R.: Incorporating Spatial Heterogeneity Created By Permafrost Thaw Into
A Landscape Carbon Estimate, J. Geophys. Res. Atmos.: Biogeosciences, 117(G1), 2012.

Belelli Marchesini, L., Papale, D., Reichstein, M., Vuichard, N., Tchebakova, N. and Valentini, R.: Carbon balance assessment of a
natural steppe of southern Siberia by multiple constraint approach, Biogeosciences, 4(4), 581–595, doi:10.5194/bg-4-581-2007,
2007.

Bell, T. W., Menzer, O., Troyo-Diéquez, E., Oechel, W. C.: Carbon Dioxide Exchange Over Multiple Temporal Scales In An Arid
Shrub Ecosystem Near La Paz, Baja California Sur, Mexico, Glob. Chang. Biol., 18(8), 2570-2582, 2012.

Beringer, J.: Adelaide River OzFlux tower site OzFlux: Australian and New Zealand Flux Research and Monitoring hdl:
102.100.100/14228, 2013a.

Beringer, J.: Dry River OzFlux tower site OzFlux, Australian and New Zealand Flux Research and Monitoring hdl:
102.100.100/14229, 2013b.

Beringer, J.:  Fogg Dam OzFlux tower site OzFlux, Australian and New Zealand Flux Research and Monitoring hdl:
102.100.100/14233, 2013c.

Beringer, J.: Sturt Plains OzFlux tower site OzFlux, Australian and New Zealand Flux Research and Monitoring hdl:
102.100.100/14230, 2013d.

Beringer, J.: Yanco JAXA OzFlux tower site OzFlux, Australian and New Zealand Flux Research and Monitoring hdl:
102.100.100/14235, 2013e.

Beringer, J.: Wallaby Creek OzFlux tower site OzFlux, Australian and New Zealand Flux Research and Monitoring hdl:
102.100.100/14231, 2013f.

Beringer, J.:  Red Dirt Melon Farm OzFlux tower site OzFlux: Australian and New Zealand Flux Research and Monitoring hdl:
102.100.100/14245, 2014a.

Beringer, J., :Riggs Creek OzFlux tower site OzFlux, Australian and New Zealand Flux Research and Monitoring hdl:
102.100.100/14246, 2014b.

Beringer, J.: Whroo OzFlux site OzFlux, Australian and New Zealand Flux Research and Monitoring hdl: 102.100.100/52559,
2017.

Biederman, J. A., Scott, R. L., Goulden, M. L., Vargas, R., Litvak, M. E., Kolb, T. E., Yepez, E. A., Oechel, W. C., Blanken, P. D., Bell,
T. W., Garatuza-Payan, J., Maurer, G. E., Dore, S., and  Burns, S. P.: Terrestrial Carbon Balance In A Drier World: The Effects Of
Water Availability In Southwestern North America, Glob. Chang. Biol., 22(5), 1867-1879, 2016.

Billesbach, D. and Arkebauer, T. J. : AmeriFlux US-SdH Nebraska SandHills Dry Valley, Dataset, doi:10.17190/AMF/1246136,
2004.

Bodesheim, J.: BACI v1, Upscaled diurnal cycles of carbon and energy fluxes. Max Planck Institute for Biogeochemistry, Jena,
(accessed on 3 October 2019),  https://doi.org/10.17871/BACI.224, 2017.

Burba, G. G. and Verma, S. B.: Prairie Growth, PAR Albedo And Seasonal Distribution Of Energy Fluxes, Agric. For. Meteorol.,
107(3), 227-240, 2001.

Burton, A. J. and Pregitzer, K. S.: Measurement Carbon Dioxide Concentration Does Not Affect Root Respiration Of Nine Tree
Species In The Field, Tree Physiology, 22(1), 67-72, 2002.

Calperum Tech; Calperum Chowilla OzFlux tower site OzFlux: Australian and New Zealand Flux Research and Monitoring hdl:
102.100.100/14236, 2013.

Campbell, J. L., Sun, O. J., and Law, B. E.: Disturbance And Net Ecosystem Production Across Three Climatically Distinct Forest
Landscapes, Global Biogeochem. Cycles, 18(4), 2004.

Castro, M. S., Gholz, H. L., Clark, K. L., Steudler, P. A.: Effects Of Forest Harvesting On Soil Methane Fluxes In Florida Slash Pine
Plantations, Canadian Journal Of Forest Research, 30(10), 1534-1542, 2000.

Cescatti, A. and Zorer, R.: Structural acclimation and radiation regime of silver fir (Abies alba Mill.) shoots along a light gradient,
Plant, Cell Environ., 26(3), 429–442, doi:10.1046/j.1365-3040.2003.00974.x, 2003.

Chen, J. M., Govind, A., Sonnentag, O., Zhang, Y., Barr, and A., Amiro, B.: Leaf Area Index Measurements At Fluxnet-Canada
Forest Sites, Agric. For. Meteorol., 140(1-4), 257-268, 2006.
Chen, X.: Surface energy balance based global land evapotranspiration (SEBS) Daily ET dataset: (accessed on 9 April 2019),
http://en.tpedatabase.cn/portal/MetaDataInfo.jsp?MetaDataId=249454, 2017.
Chi, J., Maureira, F., Waldo, S., Pressley, S. N., Stöckle, C. O., O'Keeffe, P. T., Pan, W. L., Brooks, E. S., Huggins, and D. R., Lamb, B.
K.: Carbon And Water Budgets In Multiple Wheat-Based Cropping Systems In The Inland Pacific Northwest Us: Comparison Of
Cropsyst Simulations With Eddy Covariance Measurements, Frontiers In Ecology And Evolution, 5, 25-36, 2017a.
Chi, J., Waldo, S., Pressley, S. N., Russell, E. S., O'Keeffe, P. T., Pan, W. L., Huggins, D. R., Stöckle, C. O., Brooks, E. S., and Lamb, B.
K.: Effects Of Climatic Conditions And Management Practices On Agricultural Carbon And Water Budgets In The Inland Pacific
Northwest Usa, J. Geophys. Res. Atmos.: Biogeosciences, 122(12), 3142-3160, 2017b.
Chu, H., Baldocchi, D. D., Poindexter, C., Abraha, M., Desai, A. R., Bohrer, G., Arain, M. A., Griffis, T., Blanken, P. D., O'Halloran,
T. L., Thomas, R. Q., Zhang, Q., Burns, S. P., Frank, J. M., Christian, D., Brown, S., Black, T. A., Gough, C. M., Law, B. E., Lee, X.,
Chen, J., Reed, D. E., Massman, W. J., Clark, K., Hatfield, J., Prueger, J., Bracho, R., Baker, J. M., and Martin, T. A.: Temporal
Dynamics Of Aerodynamic Canopy Height Derived From Eddy Covariance Momentum Flux Data Across North American Flux
Networks, Geophysical Research Letters, 45(5), 9275–9287, 2018.
Cleverly, J.: Ti Tree East OzFlux Site OzFlux: Australian and New Zealand Flux Research and Monitoring hdl: 102.100.100/14225,
2013.
Cleverly, J., Boulain, N., Villalobos-Vega, R., Grant, N., Faux, R., Wood, C., Cook, P. G., Yu, Q., Leigh, A., and Eamus, D.: Dy-
namics of component carbon fluxes in a semi-arid Acacia wood- land, central Australia, J. Geophys. Res.-Biogeo., 118, 1168–
1185, 2013.
Conte, M. H., Weber, J. C., Carlson, P. J., Flanagan, L. B.: Molecular And Carbon Isotopic Composition Of Leaf Wax In Vegetation
And Aerosols In A Northern Prairie Ecosystem, Oecologia, 135(1), 67-77, 2003.
Copernicus Climate Change Service (C3S): C3S ERA5-Land reanalysis. Copernicus Climate Change Service, (accessed on 11
October 2019), https://cds.climate.copernicus.eu/cdsapp#!/home, 2019.
Ershadi, A., McCabe, M. F., Evans, J. P., Chaney, N. W. and Wood, E. F.: Multi-site evaluation of terrestrial evaporation models
using FLUXNET data, Agric. For. Meteorol., 187, 46–61, doi:10.1016/j.agrformet.2013.11.008, 2014.
Eugster, W., Rouse, W. R., Pielke Sr, R. A., Mcfadden, J. P., Baldocchi, D. D., Kittel, T. G. F., Chapin, F. S., Liston, G. E., Vidale, P. L.,
Vaganov, E. and Chambers, S.: Land-atmosphere energy exchange in Arctic tundra and boreal forest: available data and
feedbacks to climate, Glob. Chang. Biol., 6(S1), 84–115, doi:10.1046/j.1365-2486.2000.06015.x, 2000.
Euskirchen, E. S., Bret-Harte, M. S., Shaver, G. R., Edgar, C. W., and Romanovsky, V. E.: Long-Term Release Of Carbon Dioxide
From Arctic Tundra Ecosystems In Alaska, Ecosystems, 20(5), 960-974, 2017.
Ewenz, C.: Loxton OzFlux tower site OzFlux: Australian and New Zealand Flux Research and Monitoring hdl: 102.100.100/20838,
2015.
Galvagno, M., Wohlfahrt, G., Cremonese, E., Rossini, M., Colombo, R., Filippa, G., Julitta, T., Manca, G., Siniscalco, C., Morra di
Cella, U. and Migliavacca, M.: Phenology and carbon dioxide source/sink strength of a subalpine grassland in response to an
exceptionally short snow season, Environ. Res. Lett., 8, 025008, doi:10.1088/1748-9326/8/2/025008, 2013.
Gash, J. H. C. and Dolman, A. J.: Sonic anemometer (co)sine response and flux measurement I. The potential for (co)sine error to
affect sonic anemometer-based flux measurements, Agric. For. Meteorol., 119, 195–207, doi:10.1016/S0168-1923(03)00137-0,
2003.
Gilmanov, T. G., Soussana, J. F., Aires, L., Allard, V., Ammann, C., Balzarolo, M., Barcza, Z., Bernhofer, C., Campbell, C. L.,
Cernusca, A., Cescatti, A., Clifton-Brown, J., Dirks, B. O. M., Dore, S., Eugster, W., Fuhrer, J., Gimeno, C., Gruenwald, T., Haszpra,
L., Hensen, A., Ibrom, A., Jacobs, A. F. G., Jones, M. B., Lanigan, G., Laurila, T., Lohila, A., G.Manca, Marcolla, B., Nagy, Z.,
Pilegaard, K., Pinter, K., Pio, C., Raschi, A., Rogiers, N., Sanz, M. J., Stefani, P., Sutton, M., Tuba, Z., Valentini, R., Williams, M. L.
and Wohlfahrt, G.: Partitioning European grassland net ecosystem CO2 exchange into gross primary productivity and ecosystem
respiration using light response function analysis, Agric. Ecosyst. Environ., 121(1–2), 93–120, doi:10.1016/j.agee.2006.12.008,
2007.
Gilmanov, T. G., Aires, L., Barcza, Z., Baron, V. S., Belelli, L., Beringer, J., Billesbach, D., Bonal, D., Bradford, J., Ceschia, E., Cook,
D., Corradi, C., Frank, A., Gianelle, D., Gimeno, C., Gruenwald, T., Guo, H., Hanan, N., Haszpra, L., Heilman, J., Jacobs, A., Jones,
M. B., Johnson, D. A., Kiely, G., Li, S., Magliulo, V., Moors, E., Nagy, Z., Nasyrov, M., Owensby, C., Pinter, K., Pio, C., Reichstein,
M., Sanz, M. J., Scott, R., Soussana, J. F., Stoy, P. C., Svejcar, T., Tuba, Z. and Zhou, G.: Productivity, Respiration, and Light-
Response Parameters of World Grassland and Agroecosystems Derived From Flux-Tower Measurements, Rangel. Ecol. Manag.,
63(1), 16–39, doi:10.2111/REM-D-09-00072.1, 2010.
Hilton, T. W., Davis, K. J. and Keller, K.: Evaluating terrestrial CO2 flux diagnoses and uncertainties from a simple land surface
model and its residuals, Biogeosciences, 11, 217–235, doi:10.5194/bg-11-217-2014, 2014.
Hirano, T., Segah, H., Harada, T., Limin, S., June, T., Hirata, R. and Osaki, M.: Carbon dioxide balance of a tropical peat swamp
forest in Kalimantan, Indonesia, Glob. Chang. Biol., 13(2), 412–425, doi:10.1111/j.1365-2486.2006.01301.x, 2007.

Hirata, R., Saigusa, N., Yamamoto, S., Ohtani, Y., Ide, R., Asanuma, J., Gamo, M., Hirano, T., Kondo, H., Kosugi, Y., Li, S. G., Nakai, Y., Takagi, K., Tani, M. and Wang, H.: Spatial distribution of carbon balance in forest ecosystems across East Asia, Agric. For. Meteorol., 148(5), 761–775, doi:10.1016/j.agrformet.2007.11.016, 2008.

Ikawa, H., Nakai, T., Busey, R., Kim, Y., Kobayashi, H., Nagai, S., Ueyama, M., Saito, K., Nagano, H., Suzuki, R. and Hinzman, L.: Understory CO2, sensible heat, and latent heat fluxes in a black spruce forest in interior Alaska, Agric. For. Meteorol., 214215, 80–90, 2015.

Irvine, J., Law, B. E. and Hibbard, K. A.: Postfire carbon pools and fluxes in semiarid ponderosa pine in Central Oregon, Glob. Chang. Biol., 13(8), 1748–1760, doi:10.1111/j.1365-2486.2007.01368.x, 2007.

King, M. D., Platnick, S., Moeller, C. C., Revercomb, H. E. and Chu, D. A.: Remote sensing of smoke, land, and clouds from the NASA ER-2 during SAFARI 2000, J. Geophys. Res. Atmos., 108(D13), doi:10.1029/2002JD003207, 2003.

Isaac P.: Daly Regrowth OzFlux tower site_old_20131128 OzFlux: Australian and New Zealand Flux Research and Monitoring hdl: 102.100.100/14215, 2010.

Jamali, H., Livesley, S. J., Dawes, T. Z., Cook, G. D., Hutley, L. B. and Arndt, S. K.: Diurnal and seasonal variations in CH4 flux from termite mounds in tropical savannas of the Northern Territory, Australia, Agric. For. Meteorol., 151(11), 1471–1479, doi:10.1016/j.agrformet.2010.06.009, 2011.

Keppel-Aleks, G., Wennberg, P. O., Washenfelder, R. A., Wunch, D., Schneider, T., Toon, G. C., Andres, R. J., Blavier, J.-F., Connor, B., Davis, K. J., Desai, A. R., Messerschmidt, J., Notholt, J., Roehl, C. M., Sherlock, V., Stephens, B. B., Vay, S. A., and Wofsy, S. C.: The imprint of surface fluxes and transport on variations in total column carbon dioxide, Biogeosciences, 9, 875–891, doi:10.5194/bg-9-875-2012, 2012.

Jung, M. et al.: FLUXCOM Global Land Energy Fluxes. Max Planck Institute for Biogeochemistry, Jena, (accessed on 2 October 2019), https://doi.org/10.17871/FLUXCOM_EnergyFluxes_v1 (2018)

Lafleur, P. M., Roulet, N. T., Bubier, J. L., Frolking, S. and Moore, T. R.: Interannual variability in the peatland-atmosphere carbon dioxide exchange at an ombrotrophic bog, Global Biogeochem. Cycles, 17(2), 1036, doi:10.1029/2002GB001983, 2003.

Laubach, J.: Beacon Farm OzFlux: Australian and New Zealand Flux Research and Monitoring hdl: 102.100.100/26730, 2016.

Li, X., Liang, S., Yu, G., Yuan, W., Cheng, X., Xia, J., Zhao, T., Feng, J., Ma, Z., Ma, M., Liu, S., Chen, J., Shao, C., Li, S., Zhang, X., Zhang, Z., Chen, S., Ohta, T., Varlagin, A., Miyata, A., Takagi, K., Saigusa, N. and Kato, T.: Estimation of gross primary production over the terrestrial ecosystems in China, Ecol. Modell., 261–262, 80–92, doi:10.1016/j.ecolmodel.2013.03.024, 2013.

Liddell, M.: Cape Tribulation OzFlux tower site OzFlux: Australian and New Zealand Flux Research and Monitoring hdl: 102.100.100/14242, 2013a

Liddell, M.: Robson Creek OzFlux tower site OzFlux: Australian and New Zealand Flux Research and Monitoring hdl: 102.100.100/14243, 2013b.

Loubet, B., Laville, P., Lehuger, S., Larmanou, E., Fléchard, C., Mascher, N., Genermont, S., Roche, R., Ferrara, R. M., Stella, P., Personne, E., Durand, B., Decuq, C., Flura, D., Masson, S., Fanucci, O., Rampon, J.-N., Siemens, J., Kindler, R., Gabrielle, B., Schrumpf, M. and Cellier, P.: Carbon, nitrogen and Greenhouse gases budgets over a four years crop rotation in northern France, Plant Soil, 343(1–2), 109–137, doi:10.1007/s11104-011-0751-9, 2011.

Macfarlane, C.: Great Western Woodlands OzFlux: Australian and New Zealand Flux Research and Monitoring hdl: 102.100.100/14226, 2013.

McCaughey, J. H., Pejam, M. R., Arain, M. A., and Cameron, D. A.: Carbon dioxide and energy fluxes from a boreal mixedwood forest ecosystem in Ontario Canada, Agr. Forest Meteorol., 140, 79–96, 2006.

Mu, Q.: Mod16a2_monthly.Merra_gmao_1kmalb. (accessed on 1 October 2019), http://files.ntsg.umt.edu/data/NTSG_Products/MOD16/MOD16A2_MONTHLY.MERRA_GMAO_1kmALB/, 2015.

Noormets, A., Chen, J. and Crow, T. R.: Age-Dependent Changes in Ecosystem Carbon Fluxes in Managed Forests in Northern Wisconsin, USA, Ecosystems, 10(2), 187–203, doi:10.1007/s10021-007-9018-y, 2007.

ORNL DAAC: Home | fluxnetweb.ornl.gov, [online] Available from: https://fluxnet.ornl.gov/ (Accessed 1 July 2019), 2015.

Pendall E.: Cumberland Plain OzFlux Tower Site OzFlux: Australian and New Zealand Flux Research and Monitoring hdl: 102.100.100/25164, 2015.

Saleska, S.R., H.R. da Rocha, A.R. Huete, A.D. Nobre, P. Artaxo, and Y.E. Shimabukuro: . LBA-ECO CD-32 Flux Tower Network Data Compilation, Brazilian Amazon: 1999-2006. Data set. Available on-line [http://daac.ornl.gov] from Oak Ridge National Laboratory Distributed Active Archive Center, Oak Ridge, Tennessee, USA, doi:10.3334/ORNLDAAC/1174, 2013.

Reichstein, M., Rey, A., Freibauer, A., Tenhunen, J., Valentini, R., Banza, J., Casals, P., Cheng, Y., Grünzweig, J. M., Irvine, J., Joffre, R., Law, B. E., Loustau, D., Miglietta, F., Oechel, W., Ourcival, J.-M., Pereira, J. S., Peressotti, A., Ponti, F., Qi, Y., Rambal, S., Rayment, M., Romanya, J., Rossi, F., Tedeschi, V., Tirone, G., Xu, M. and Yakir, D.: Modeling temporal and large-scale spatial variability of soil respiration from soil water availability, temperature and vegetation productivity indices, Global Biogeochem. Cycles, 17(4), doi:10.1029/2003GB002035, 2003.

Reichstein, M., Falge, E., Baldocchi, D., Papale, D., Aubinet, M., Berbigier, P., Bernhofer, C., Buchmann, N., Gilmanov, T., Granier, A., Grunwald, T., Havrankova, K., Ilvesniemi, H., Janous, D., Knohl, A., Laurila, T., Lohila, A., Loustau, D., Matteucci, G., Meyers, T., Miglietta, F., Ourcival, J.-M., Pumpanen, J., Rambal, S., Rotenberg, E., Sanz, M., Tenhunen, J., Seufert, G., Vaccari, F., Vesala,

T., Yakir, D. and Valentini, R.: On the separation of net ecosystem exchange into assimilation and ecosystem respiration: review
and improved algorithm, Glob. Chang. Biol., 11(9), 1424–1439, doi:10.1111/j.1365-2486.2005.001002.x, 2005.
Revill, A., Sus, O., Barrett, B. and Williams, M.: Carbon cycling of European croplands: A framework for the assimilation of
optical and microwave Earth observation data, Remote Sens. Environ., 137, 84–93, doi:10.1016/j.rse.2013.06.002, 2013.
Schroder, I.:  Arcturus Emerald OzFlux tower site OzFlux: Australian and New Zealand Flux Research and Monitoring hdl:
102.100.100/14249, 2014.
Silberstein, R: Gingin OzFlux, Australian and New Zealand Flux Research and Monitoring hdl: 102.100.100/22677, 2015.
Soegaard, H. and Nordstroem, C.: Carbon dioxide exchange in a high-arctic fen estimated by eddy covariance measurements
and modelling, Glob. Chang. Biol., 5(5), 547–562, doi:10.1111/j.1365-2486.1999.00250.x, 1999.
Stoy, P. C., Mauder, M., Foken, T., Marcolla, B., Boegh, E., Ibrom, A., Arain, M. A., Arneth, A., Aurela, M., Bernhofer, C., Cescatti,
A., Dellwik, E., Duce, P., Gianelle, D., van Gorsel, E., Kiely, G., Knohl, A., Margolis, H., Mccaughey, H., Merbold, L., Montagnani,
L., Papale, D., Reichstein, M., Saunders, M., Serrano-Ortiz, P., Sottocornola, M., Spano, D., Vaccari, F. and Varlagin, A.: A data-
driven analysis of energy balance closure across FLUXNET research sites: The role of landscape scale heterogeneity, Agric. For.
Meteorol., 171–172, 137–152, doi:10.1016/j.agrformet.2012.11.004, 2013.
Stefan A.: Wombat State Forest OzFlux-tower site OzFlux: Australian and New Zealand Flux Research and Monitoring hdl:
102.100.100/14237, 2013.
Sulkava, M., Luyssaert, S., Zaehle, S. and Papale, D.: Assessing and improving the representativeness of monitoring networks:
The European flux tower network example, J. Geophys. Res., 116(G3), G00J04, doi:10.1029/2010JG001562, 2011.
Wei, S., Yi, C., Hendrey, G., Eaton, T., Rustic, G., Wang, S., Liu, H., Krakauer, N. Y., Wang, W., Desai, A. R., Montagnani, L., Tha
Paw U, K., Falk, M., Black, A., Bernhofer, C., Grünwald, T., Laurila, T., Cescatti, A., Moors, E., Bracho, R. and Valentini, R.: Data-
based perfect-deficit approach to understanding climate extremes and forest carbon assimilation capacity, Environ. Res. Lett.,
9, 065002, doi:10.1088/1748-9326/9/6/065002, 2014.
Wood, E.: Princeton University SRB/GEWEX evapotranspiration (Penman-Monteith) L4 3 hour 0.5 degree x 0.5
degree V1, Greenbelt, MD USA, Goddard Earth Sciences Data and Information Services Center (GES DISC),
(accessed on 1 October 2019), 10.5067/MEASURES/WATERCYCLE/DATA314, 2017
Woodgate, W.: Tumbarumba OzFlux tower site OzFlux: Australian and New Zealand Flux Research and Monitoring hdl:
102.100.100/14241, 2013.
Zhang, Y., Pena A. J., McVicar, T. Chiew, F.; Vaze, J., Zheng, H., Wang, Y.: Monthly global observation-driven Penman-Monteith-
Leuning (PML) evapotranspiration and components. v2. CSIRO. Data Collection. (accessed on 9 April 2019),
https://doi.org/10.4225/08/5719A5C48DB85, 2016.
Zhang, Ke: PLSH Monthly Global 8kmResolution: (accessed on 30 September 2019),
http://files.ntsg.umt.edu/data/ET_global_monthly/Global_8kmResolution/, 2017.

---

## Author Response (AR2)

We would like to thank the editor and the referees for their constructive comments on our manuscript. This document outlines our responses to the final revision. Below we highlight the changes that we made to the manuscript.

**Response to Editor**

Referee #1 has looked at your revision, and he is satisfied with the latest version. However he pointed out two minor issues that you might want to look at:

- Table 5: I suggest the authors compute the trends over the same periods across all parent data sets for a fairer comparison.

We have now computed the trends over a common period 1982 – 2012, and updated the table and the text that explains the results in Lines 664 - 673.

*Table 5: Trends in yearly ET total (mm year$^{-1}$) spatially averaged across each ET regime calculated for DOLCE V3 and five participating parent datasets available during 1982 – 2012. The text shows slopes of the trend line and their confidence interval calculated at the 95% confidence level, bold text indicates that the trend is reliable since the confidence interval is strictly positive or negative.*

| Dataset and time span | V.L.ET, H.variability | L.ET, H.variability | M.L.ET, M.variability | M.H.ET, M.variability | H.ET, L.variability | V.H.ET, L.variability |
|---|---|---|---|---|---|---|
| **DOLCE V3** | -0.04 [-0.23, 0.16] | 0.26 [-0.11, 0.63] | **0.44 [0.1, 0.76]** | **0.56 [0.2, 0.87]** | 0.07 [-0.27, 0.4] | 0.34 [-0.1, 0.9] |
| **ERA5-land** | -0.18 [-0.36, 0.04] | 0.02 [-0.42, 0.47] | 0.14 [-0.38, 0.6] | **-0.65 [-1.14, -0.22]** | **-0.89 [-1.28, -0.51]** | 0.11 [-0.2, 0.5] |
| **FLUXCOM-MET** | **-0.02 [-0.04, 0]** | 0.04 [-0.11, 0.23] | 0.05 [-0.07, 0.2] | -0.11 [-0.27, 0.04] | -0.003 [-0.18, 0.17] | 0.25 [-0.04, 0.57] |
| **GLEAM 3.5A** | -0.08 [-0.28, 0.16] | 0.35 [-0.04, 0.76] | **0.59 [0.34, 0.95]** | **0.43 [0.1, 0.77]** | 0.05 [-0.33, 0.44] | **0.62 [0.12, 1.31]** |
| **PML** | -0.1 [-0.28, 0.15] | **0.42 [0.11, 0.75]** | **1 [0.64, 1.45]** | 0.21 [-0.19, 0.64] | 0.28 [-0.38, 0.81] | -0.32 [-1.24, 0.62] |
| **PLSH** | **0.17 [0.1, 0.24]** | **0.39 [0.16, 0.66]** | **1.3 [0.8, 1.77]** | **1.41 [0.85, 1.89]** | **1.53 [0.75, 2.17]** | **0.82 [0.36, 1.35]** |

*We repeat the same analysis for all the participating parent datasets that span at least 30 years. Sen's slope of the trends over the period 1982 – 2012 and their confidence interval (computed at the 95% confidence level) are presented in Table 5. As noted earlier, trends' behaviour is deemed inconclusive when the CI encompasses negative and positive values. These are presented with regular (as opposed to bold) typeface and are exhibited by FLUXCOM-MET in all regimes except the driest. In contrast, PLSH shows reliable upward trends in all regimes. ERA5-land shows downward trends in the 'M.H.ET, M.variability' and 'H.ET, L.variability' regimes. Both GLEAM 3.5A and DOLCE V3 show reliable upward ET trends in the two middle regimes. Differences exist in the magnitude of*

*trends across the majority the products and the regimes. In DOCLE V3, the strongest trend occur in the 'M.H.ET, M.variability' regime at a rate 0.56 $mm\ year^{-1}$. Finally, the slopes of DOLCE V3 trends are within the range of slopes of trends in available ET products.*

- L98: "direst" --> "driest"

We thank the referee for spotting this. We have now made the change in the text.